psychology/neuroscience

language lateralization, hemispheric dominance, functional transcranial Doppler sonography, structural equation modelling, speech production, speech comprehension

**Author for correspondence:**
Z. V. J. Woodhead
e-mail: zoe.woodhead@psy.ox.ac.uk

# Testing the unitary theory of language lateralization using functional transcranial Doppler sonography in adults

Z. V. J. Woodhead, A. R. Bradshaw, A. C. Wilson,
P. A. Thompson and D. V. M. Bishop

Department of Experimental Psychology, University of Oxford, Oxford, UK

(iD) ZVJW, 0000-0003-0462-6791

Hemispheric dominance for language can vary from task to task, but it is unclear if this reflects error of measurement or independent lateralization of different language systems. We used functional transcranial Doppler sonography to assess language lateralization within the middle cerebral artery territory in 37 adults (seven left-handers) on six tasks, each given on two occasions. Tasks taxed different aspects of language function. A pre-registered structural equation analysis was used to compare models of means and covariances. For most people, a single lateralized factor explained most of the covariance between tasks. A minority, however, showed dissociation of asymmetry, giving a second factor. This was mostly derived from a receptive task, which was highly reliable but not lateralized. The results suggest that variation in the strength of language lateralization reflects true individual differences and not just error of measurement. The inclusion of several tasks in a laterality battery makes it easier to detect cases of atypical asymmetry.

## 1. Introduction

Hemispheric dominance for language is often assumed to be unidimensional and consistent across language domains, but this assumption can be questioned [1,2]. Discrepant laterality across different language tasks [3–5] could be simply due to measurement error [6]; alternatively, task differences may represent meaningful individual variation in the hemispheric organization of different language networks. It has been difficult to distinguish these possibilities, because, while we have ample evidence that the left hemisphere is heavily implicated in language function at the group level, relatively little is known about the reliability of lateralization in individuals. It is evident

that a standard model based on average brain activation may give a misleading impression of uniformity [7]. Furthermore, there is evidence that there may be subgroups of people with distinct laterality profiles, related to handedness [8]. Such variability in cerebral lateralization may have functional significance, for example, in terms of impaired language abilities [1]. In clinical neurosurgical contexts, it is important to know whether a single indicator of an individual's language laterality is sufficient, or whether a battery of measures is needed to capture laterality in multiple language domains [3–5]. Before we can make headway in answering such questions, we need to have reliable measures.

Here, we report a study using functional transcranial Doppler sonography (fTCD) [9] to measure the speed of blood flow in left and right middle cerebral arteries (a proxy for neural activity in language-related areas of the brain) during six different language tasks (tasks A–F). The tasks were designed to assess lateralization across a broad range of language functions (namely phonology, semantics, syntax, speech production and speech perception), while keeping non-linguistic demands as closely matched as possible. The fTCD data were used to derive laterality indices (LIs), which quantify the balance of activation in the middle cerebral artery (MCA) territories of the left and right hemispheres. All participants were tested on the whole battery in two separate sessions on different days in order to estimate the reliability of the LIs and the extent to which lateralization of different tasks could be explained in terms of a common factor.

## 1.1. Laterality at the level of the population and the individual

The question of whether language lateralization is a unitary function has two distinct interpretations: (a) whether there are differences in the extent of lateralization across different language functions or (b) whether there are individual differences in how the strength of lateralization varies across language functions. We first review existing literature on these questions and then present simulated data to show how predictions made by the two accounts are independent and additive, but can be tested within a common framework (structural equation modelling, SEM).

## 1.2. Task-related variation in the extent of language lateralization

Most theories of language lateralization have focused on how language functions are lateralized in the brain in typical humans. Such theories are not concerned with individual differences, but make theoretical statements about the properties of language that are associated with lateralized activity. An influential example of such a theory is Hickok and Poeppel's dual route model of speech processing [10]. This contrasts a dorsal stream from superior temporal to premotor cortices via the arcuate fasciculus, which is associated with sensorimotor integration of auditory speech sounds and articulatory motor actions; and a ventral stream from temporal cortex to anterior inferior frontal gyrus, which is involved in access to conceptual memory and mapping of sound to meaning [11]. Hickok and Poeppel proposed that the dorsal stream is left lateralized, whereas the ventral stream is bilateral. This kind of theory makes predictions about task-related differences that can be assessed by comparing mean LIs in a sample. Thus, the prediction from the dual route model is that mean LIs for tasks involving the dorsal stream will show left lateralization, whereas LIs from tasks primarily involving the ventral stream will not be lateralized.

Hickok and Poeppel's model contrasts with other theoretical accounts. For instance, Dhanjal et al. proposed that left lateralization was a characteristic of tasks involving lexical retrieval [12]. Evidence came from an fMRI study investigating propositional speech (e.g. sentence generation) and non-propositional speech (e.g. reciting memorized speech): articulatory jaw and tongue movements and non-propositional speech co-activated bilateral dorsal areas, including the superior temporal planes, motor and premotor cortices. Only the lexical retrieval component of propositional speech resulted in left-lateralized activity (in the inferior frontal gyrus and premotor cortex).

Yet other accounts have focused on the complexity of the speech stimulus [13], or argued that lateralization is specifically linked to aspects of complex syntactic processing [14,15].

## 1.3. Individual differences in cerebral lateralization

Discussions about the nature of language lateralization are complicated by individual differences; although most people show the typical pattern of language laterality, some individuals show the reverse pattern—right-hemisphere language. In a large-scale comparison of left- and right-handers, Mazoyer et al. [8] reported that strong right-hemisphere bias for a sentence generation task was seen exclusively in left-handers, though milder departures from left hemisphere dominance were seen in

right- as well as left-handers. A subset of people with bilateral language has also been described for many years [16], but this category is ambiguous. These could be people who engage both hemispheres equally during language tasks, or people who are strongly lateralized for different tasks, but in different directions. This latter scenario would provide strong evidence against a unitary hypothesis, by demonstrating that a person's language laterality could not be predicted by a single dimension.

Individual differences in cerebral lateralization have previously been observed in the comparison between left-lateralized verbal functions versus right-lateralized nonverbal functions. This might suggest complementarity of the two functions within the brain; however, where individual differences in these biases have been assessed, several studies have found them to be dissociated ([17–21], cf. [22]). Again, handedness has been noted as an important factor, with right-handers showing less evidence of complementarity of verbal and visuospatial functions than left-handers [21]. Here, we consider whether similar dissociations might be found *within* the domain of language. Although previous investigators have considered association or dissociation in average patterns of activation for different tasks [23,24], there has been little previous research documenting individual differences in task-related variation. Inconsistent LIs from task to task could simply reflect noisy measurement, making dissociations hard to interpret. Thus, in order to throw light on individual differences in language laterality, we need to include repeated measures, so that the reliability of LIs from different tasks can be assessed.

## 1.4. Simulated data to illustrate predictions

It is possible to integrate models of task variation in lateralization with a model of individual differences in the kind of framework shown in figure 1. The script used to generate this simulated data can be found on Open Science Framework (OSF; https://osf.io/dbm4p/). For simplicity, this shows simulated data on just two tasks, A and B, to contrast predictions from different models of the structure of language lateralization. The Population Bias model is the simplest: it shows a population bias to left-sided language laterality (i.e. positive LI values) that does not depend on the task. There are no consistent individual differences: any variation in laterality is just caused by random error. This is not a very plausible model, but provides a useful starting point from which to build more complex scenarios. Formally, the function of predicting an individual's LI is as follows:

$$\mathrm{LI}_{ij} = a + e_{ij}$$

where $i$ indexes the task, and $j$ the individual, $a$ is an intercept term corresponding to population bias, and $e$ is random error.

In the Population Bias model, the mean LIs for different language tasks (shown by the horizontal and vertical red dotted lines) are all the same and equal to $a$ (in this case set to 1). Note that because there are no stable individual differences, the correlations between LIs for the same task measured on different occasions (left-hand panel), and between different tasks measured on the same occasion (right-hand panel) are zero.

The second model is the Task Effect model. This incorporates consistent task-specific variation, without any stable individual differences. Formally,

$$\mathrm{LI}_{ij} = a + t_i + e_{ij}$$

where $t_i$ is a task-specific term. The only difference from the Population Bias model is that the means differ for different tasks—i.e. tasks A and B have mean LIs of 1 and 2, respectively. Again, variation in individuals' LI scores is due to random error ($e$), rather than any systematic individual differences, as evidenced by zero test–retest correlations.

The next model is a Person Effect model. This includes stable individual differences: a person's score on any test occasion depends on an intrinsic lateral bias, which is constant from task to task but varies from person to person, i.e.

$$\mathrm{LI}_{ij} = a + t_i + p_j + e_{ij}$$

where $p_j$ is the person-specific term. This model predicts significant correlations between the same task tested on different occasions, and different tasks tested on the same occasion. An important point is that these correlations depend solely on the relative contribution of individual difference ($p$) versus random noise ($e$) to the LI. It does not matter whether there are also task-related effects ($t$) on the LI. Thus, in the example, we have one task that is lateralized (mean LI of 2) and one that is not (mean LI of 0), yet on this model, the test–retest correlation for either task will be the same, and equivalent to the cross-task correlation.

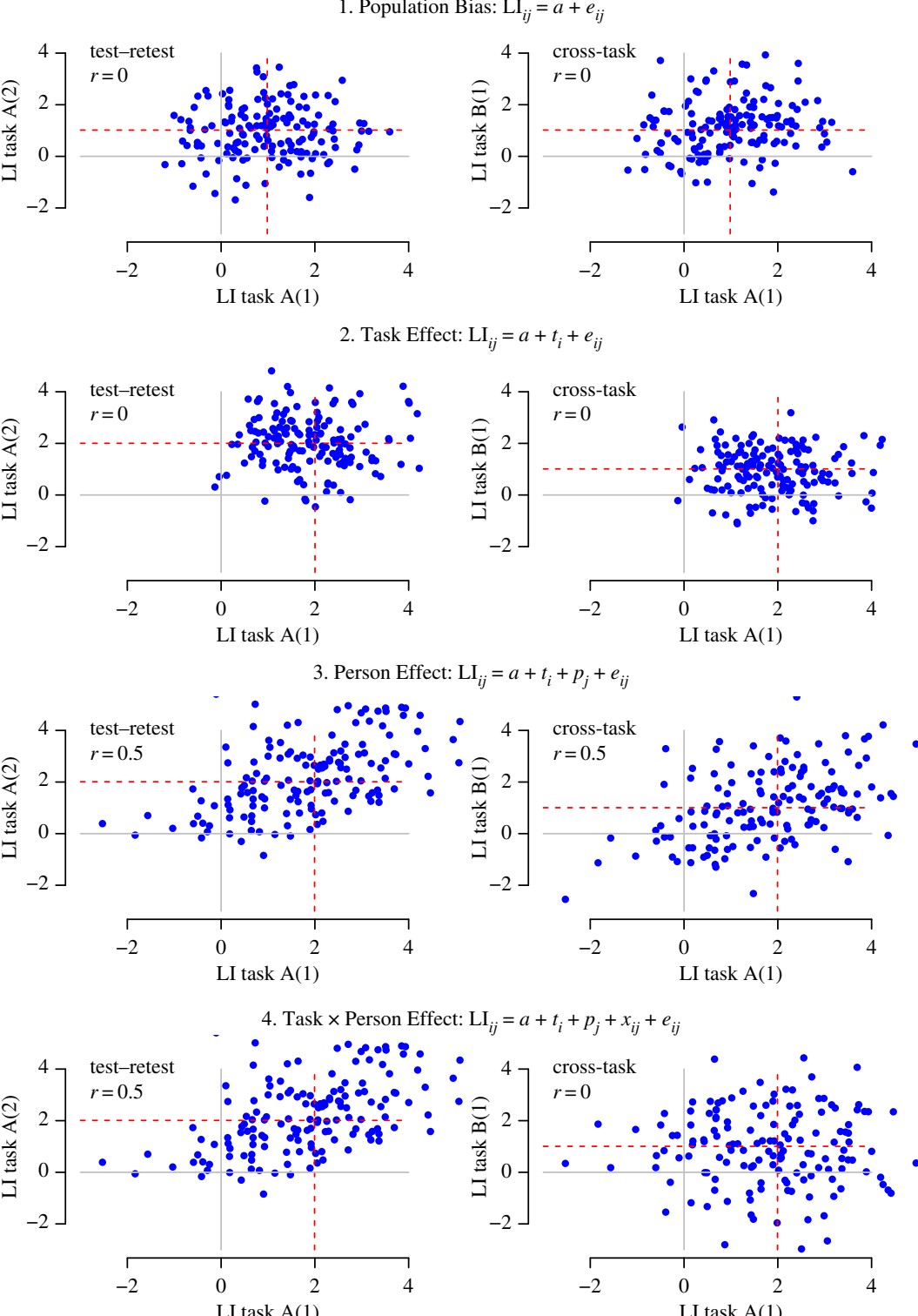

**Figure 1.** Simulated data of different theoretical models of variance across sessions (1 and 2) and tasks (A and B) in language lateralization. Red dotted lines show the mean lateralization index (LI) for the task/session. For further details, see https://osf.io/dbm4p/.

The final model incorporates a Task by Person Effect: i.e. there are stable individual differences that show up as significant test–retest reliability on any one task, but the rank ordering of lateralization varies from task to task, so cross-task correlations are low. Formally,

$$LI_{ij} = a + t_i + p_j + x_{ij} + e_{ij}$$

where $x_{ij}$ reflects a contribution that is specific to the task and the individual. The depicted scenario in figure 1 is an extreme one, with no relationship between a person's laterality on tasks A and B; in practice, there could be significant cross-task correlations, but if the within-task correlations are higher than cross-task correlations, then this would be evidence that individual differences in laterality are to some extent task-specific.

A key point illustrated by these simulations is that testing the multivariate model of language laterality at the population level requires different evidence—i.e. testing between means—than a multivariate model of individual differences, which requires us to consider correlations within and between tasks. Furthermore, predictions from these two types of model are independent, because correlations are not influenced by mean values.

We used SEM to evaluate the relative fit of these four models to data on language lateralization for participants who have LIs assessed on a range of tasks on two occasions. SEM was used for a number of reasons, including: (i) it allowed us to model the unobserved factors (also known as latent variables or constructs) predicting lateralization strength across the observed tasks and sessions (also known as manifest variables or indicators), (ii) it modelled both the mean strength of lateralization for each task as well as the covariance between tasks, (iii) it explicitly modelled the residuals associated with each latent variable, which allowed measurement error to be accounted for, and (iv) it allowed for different models to be compared directly using likelihood ratio hypothesis testing.

## 1.5. Hypotheses

We pre-registered a set of hypotheses that were tested through SEM model comparison, as described in the Methods.

We first tested two hypotheses concerning the group mean LI values. First, we tested the dorsal stream hypothesis [10], which predicts that strength of lateralization depends on the extent to which tasks map on to the dorsal versus ventral speech processing streams (dorsal = stronger left lateralization). Second, following Dhanjal *et al.* [12], we tested the lexical retrieval hypothesis, which maintains that lateralization depends on the extent to which tasks require lexical retrieval (more lexical retrieval = stronger left lateralization).

A second set of hypotheses concerned individual differences in LI value. We predicted that a Task by Person Effect model, whereby covariances between tasks were modelled by two latent factors, would give a better fit to the data than a Person Effect model, where covariances were modelled by only one factor.

Our approach was to use SEM to test a series of pre-specified, hypothesis-driven models. This differs from using SEM to identify an optimal fit to the data via *post hoc* model modifications. We fitted all models without modification from our pre-registered analysis plan and, to this end, we have reported the fit of every model, even suboptimal ones, as the important detail is whether they offer an improved fit relative to previous models in the series. This approach is of particular importance in designs with (relatively) small sample sizes and low degrees of freedom, where overfitting is a real concern.

# 2. Methods

## 2.1. Pre-registration and data sharing

This project was pre-registered on OSF prior to data collection, and all task materials, analysis scripts and anonymized data can also be found on OSF (https://osf.io/tkpm2/). A number of changes were made to the analysis plan after collection of the data—an updated protocol is documented here: https://osf.io/bjsv8/. Departures from the original protocol are explained in the Departures from pre-registered methods section.

## 2.2. Design

A test−retest, within-subject design was used. Lateralization of brain activity was measured using functional transcranial Doppler sonography (fTCD) during six language tasks: (A) List Generation, (B) Phonological Decision, (C) Semantic Decision, (D) Sentence Generation, (E) Sentence Comprehension and (F) Syntactic Decision. Participants were tested on two sessions spaced by between 3 days and 6 weeks. Hence, each participant provided data from six tasks tested twice (A1-F1, A2-F2).

## 2.3. Participants

A sample size of $n = 30$ was determined by simulations of data from six tasks administered on two occasions, to determine the smallest sample size that would reliably distinguish data generated from a two factor versus single factor model, and give acceptable fit indices (see SEM_power_test.R, https://osf.io/r5e3p/). The simulations were based on the models of covariances, as the factor structure of the measures is our primary interest, and this gave a more conservative power estimate. We note that the sample size is small relative to those usually recruited for SEM analyses. However, because all measures were taken twice, with no practice effects expected (on the basis of previous studies with this method), there are several estimates of most parameters. For instance, the correlation between LIs for tasks A and B is estimated from A1B1, A1B2 and A2B2. Thus, the repeated measures give low degrees of freedom relative to the number of measures.

In our original study pre-registration, we did not plan to select participants according to handedness. However, both prior literature and our own preliminary data indicated that it would be advisable to treat right- and left-handers separately, as the pattern of associations between language tasks appeared to differ according to handedness, so combining handedness groups could give a misleading picture. We became concerned that results from our pre-registered analysis on 30 participants (seven left-handers) were potentially misleading, as the factor structure that emerged seemed driven by a few left-handers. We, therefore, tested additional participants to give a total sample of 30 right-handers and seven left-handers, and we report analysis based on this larger sample as exploratory results.

All participants gave written informed consent. Procedures were approved by the University of Oxford's Medical Sciences Interdivisional Research Ethics Committee (approval number R40410/RE004). Subjects were recruited using the Oxford Psychology Research Participant Recruitment Scheme (https://opr.sona-systems.com) and by poster advertisements. The inclusion criteria were: aged 18–45 years; English native language speakers; and with normal or corrected to normal hearing and vision. Exclusion criteria were: a history of significant neurological disease or head injury; or a history of developmental language disorder.

It was not possible to record a Doppler signal via the temporal window in three participants. In these cases, the participant was reimbursed but not tested further, and another participant was recruited in their place. One participant had excessive motion artefacts in his first session, so another participant was recruited in his place. The initial group of 30 participants (17 female and seven left-handed) had a mean age of 26.0 years (s.d. = 7.2 years; range: 19.2–45.1 years). The final group, including seven additional right-handers (two females) had mean age 25.9 years (s.d. = 6.8 years) with the same age range.

## 2.4. Procedure

The order of the six language tasks was counterbalanced between subject and session. At each session, 15 trials of each task type were conducted with breaks in between tasks.

## 2.5. Language tasks

The six tasks were designed to be matched in trial structure, as far as feasible, so that differences in laterality should reflect as far as possible the linguistic task demands. The number of trials per condition was the same for all tasks. We manipulated the task timings so that trial lengths (and therefore the time on task and time since baseline) would be the same for all conditions. We were able to make the timings within a trial for the phonological decision, semantic decision and sentence comprehension tasks identical. Pilot testing was carried out and difficulty levels were titrated to make task performance (accuracy and reaction time (RT) for decision tasks; the number of words spoken for speech production tasks) as similar as possible between tasks. The first five tasks had a visual stimulus on each trial presented against a grey background, to keep the visual demands as similar as possible; the sixth task involved presentation of written words. All stimulus materials are available on OSF (https://osf.io/8s7vn/).

The rest period prior to stimulus presentation was used for baseline correction to equate the left and right channels. Trials were 33 s long, and followed the structure shown in figure 2. Trials started with the word 'CLEAR' on screen for 3 s, indicating that participants must clear their mind in preparation for the next trial. The language task followed, lasting for 20 s. Procedures for each task type are detailed below, and examples of stimuli are shown in figure 3. Note that for tasks B, C, E and F, participants made

| | 0 | 3 | 6 | 17 | 23 | 33 s |
|---|---|---|---|---|---|---|
| A. List generation | clear mind | stimulus | list generation | | report | rest |
| B. Phonological decision | clear mind | | phonological decision × 6 | | | rest |
| C. Semantic decision | clear mind | | semantic decision × 6 | | | rest |
| D. Sentence generation | clear mind | stimulus | sentence generation | | report | rest |
| E. Sentence comprehension | clear mind | | sentence decision × 6 | | | rest |
| F. Syntactic decision | clear mind | | syntactic decision × 3 | | | rest |

**Figure 2.** Timings within a single trial for all six task types.

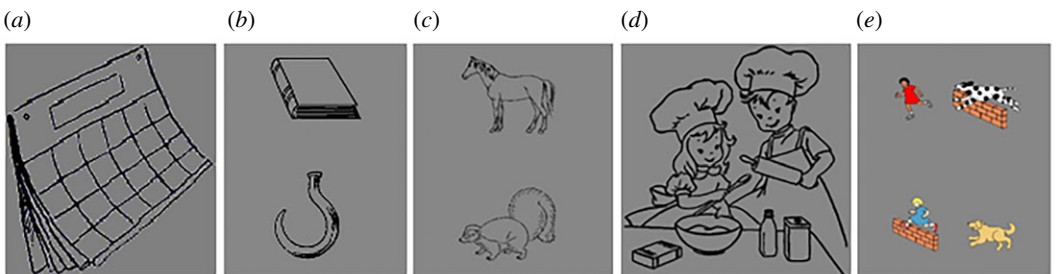

*(a)* *(b)* *(c)* *(d)* *(e)*

**Figure 3.** Example stimuli for the language tasks. From left to right: picture stimulus for List Generation task (*a*; recite months of the year); a matching picture pair (book/hook) for the Phonological Decision task (*b*); a matching picture pair for the Semantic Decision task (*c*); picture stimuli for the Sentence Generation task (*d*); and a picture pair for the Sentence Comprehension task (*e*; The dog chases the girl who is jumping).

responses to a series of stimuli on each trial to ensure the participant was engaged in language processing throughout the activation interval. Rapid presentation of multiple stimuli in a trial has been shown by Payne *et al.* [25] to maximize lateralized activation in fTCD. After the task, 'REST' appeared on screen for 10 s, during which participants were required to clear their minds.

### 2.5.1. List generation

This task was based on the reference task used by Mazoyer *et al.* [8]. Participants were asked to recite an automatic sequence of words (non-propositional speech) in response to a picture. In each trial, a line drawing was displayed on a grey background for 3 s. Participants were trained to produce different sequences for different pictures: reciting the numbers from 1 to 10, the letters from A–J, the days of the week or the months of the year. A fixation cross was then presented in the centre of the screen for 11 s, during which the participant recited the words covertly (silently) in their head. Following this, a 'REPORT' prompt was shown for 6 s, indicating that participants should say the sequence aloud.

Similar tasks in fMRI [12] and fTCD [26] showed bilateral activations. The list generation task involves generation of phonological output, and so should index the dorsal stream, but because it involves repeated, overlearned material, it does not implicate the ventral stream; nor does it place demands on lexical retrieval. Thus, the two specific theories of interest make contrasting predictions about this task.

### 2.5.2. Phonological decision

Participants were required to make a rhyme judgement on pairs of words represented by pictures. The pictures were easily nameable line drawings of single syllable words, mostly taken from the International Picture Naming Project (IPNP) database (https://crl.ucsd.edu/experiments/ipnp/index.html) [27]. The pictures were arranged into 45 rhyming and 45 non-rhyming pairs (based on pairings devised by Bishop & Robson) [28]. Rhyming and non-rhyming pairs did not differ significantly on orthographic similarity (assessed using MatchCalc software, http://www.pc.rhul.ac.

uk/staff/c.davis/Utilities/MatchCalc/). For each trial, a series of six picture pairs were presented, each for 3.33 s (totalling 20 s). For each pair, the participant decided whether the words represented by the pictures rhymed or not, and responded by button press.

Similar rhyme decision tasks in fMRI [29] and fTCD [25] elicited left lateralization. This task involves implicit generation of lexical items and their phonology, but does not require access to conceptual meaning. Both the dorsal–ventral stream theory and lexical retrieval theory predict it should be strongly lateralized.

### 2.5.3. Semantic decision

This task involved a semantic category judgement on objects represented in a pair of pictures.

The design of this task closely matched that of the phonological decision task. The pictures were mostly taken from the IPNP database, as described above. The stimuli were matched for word familiarity, orthographic neighbourhood, imageability, number of phonemes and frequency. Six picture pairs were presented, each for 3.33 s. For each pair, the participant decided whether the objects were from the same semantic category or not (e.g. both types of food) and responded by button press.

In fMRI, similar semantic decision tasks elicit left lateralization of ventral language areas [30,31], and in fTCD it has shown to have moderate and highly reliable left lateralization [4,32]. For this task, it is necessary to access conceptual meaning, but generation of word names is not implicated. This, then, can be regarded as indexing the ventral stream. Both the dorsal–ventral stream theory and the lexical retrieval theory predict weak lateralization for this task.

### 2.5.4. Sentence generation

This task required participants to generate spoken sentences in response to line drawings, following methods described by Mazoyer *et al.* [8], but using pictures that were more culturally appropriate for UK participants.

For each trial, a black line drawing was displayed on a grey background for 3 s. This was followed by a fixation cross for 11 s, during which the participant was required to covertly generate a sentence. Participants were trained in advance to generate sentences beginning with a subject (e.g. 'the boy'), followed by a description of the subject (with marbles), a verb (plays) and ending with a detail about the action (on the floor). A 'REPORT' prompt was then presented for 6 s, and participants were required to say their sentence aloud.

In fTCD, this task has been shown to be strongly left lateralized [26]. In fMRI, picture description has been shown to activate dorsal stream areas including the posterior temporo-parietal cortex, premotor cortex and inferior frontal cortex; but also ventral regions such as posterior and anterior portions of the middle temporal gyrus [8,33]. These activations were bilateral, but biased to the left. This task implicates both dorsal and ventral streams, and so might be expected to show weaker lateralization than purely dorsal tasks. By contrast, the lexical retrieval theory predicts strong lateralization.

### 2.5.5. Sentence comprehension

This task required participants to decide which of two pictures corresponded to a spoken sentence. Each trial comprised six picture pairs, each presented for 3.33 s, along with a spoken sentence that matched one of the two pictures. The sentences were spoken at a rapid pace and included some involving complex grammar with long-distance dependencies, such as 'the shoe on the pencil is blue', or 'the cow that is brown is chasing the cat'. Participants indicated which of the two pictures matched the sentence by button press.

This task would appear to stress the ventral more than the dorsal stream, and so might be predicted to have relatively weak lateralization. Auditory comprehension tasks have previously been shown to activate left-lateralized ventral language areas [34,35] and have shown moderate left lateralization with fTCD [36]. The task is hard to categorize in terms of lexical retrieval: it is necessary to hold word meanings in memory while working out the meaning, though overt word generation is not required.

### 2.5.6. Syntactic decision

This task was designed to isolate syntactic processing with minimal involvement of semantics. This task uses 'Jabberwocky' stimuli, based on a study by Fedorenko *et al.* [37], where content words of sentences are replaced by plausible non-words. Half of the stimuli were 'sentences', where function words, word

order and morphological cues were preserved to make the stimuli recognizable as syntactically valid sentences (e.g. The tarben yipped a lev near the kruss). The other half had a pseudorandom word order and were not perceived as sentences (e.g. Kivs his porla her tal ghep in with).

Each trial contained three Jabberwocky stimuli of eight words. Words were presented sequentially at the same time as an audio recording of the spoken word. As all spoken words were recorded separately, there were no prosodic cues to whether the stimulus is a 'sentence' or not. Each word was presented for 0.7 s, and the sequence was followed by a question mark for 1 s (making a total of 6.7 s for each Jabberwocky stimulus). The participant was required to respond by button press following the '?' prompt to indicate whether they thought the sequence formed a sentence or not.

An fMRI study using Jabberwocky sentences [38] showed small areas of left posterior superior temporal and inferior frontal gyri that were sensitive to the syntactic structure of sentences in the absence of meaningful semantics. In terms of the dorsal–ventral stream account, this task is predicted not to show lateralization, as it is a purely receptive task. This was the only task involving non-words, and should not be lateralized according to a lexical retrieval account.

## 2.6. Behavioural analysis

For tasks A and D, the average number of words generated for each trial was calculated. For tasks B, C, E and F, percentage accuracy and average reaction time for correct trials (excluding trials where reaction time was greater than 2 s.d. away from the mean) were calculated. The number of events where no response was received was also recorded for each task—these events were scored as incorrect.

## 2.7. fTCD analysis

Our analysis of fTCD data departed from the method we pre-registered in three respects; sections describing the altered methods are shown in italics, with a description and explanation of the change shown in the section 'Departures from pre-registered methods'.

The dependent measures derived from the fTCD analysis were the laterality indices (LI) from tasks A to F at sessions 1 and 2. fTCD uses ultrasound probes positioned bilaterally over the temporal windows to measure cerebral blood flow velocity (CBFV) in the left and right MCA. The probes emit ultrasound pulses and detect reflected ultrasound signal. The frequency of the reflected ultrasound signal depends on the speed of the blood moving in the MCA, due to Doppler shift. Hence the difference in frequency of the emitted and reflected ultrasound signals can be used to determine the speed of blood flow. The data are recorded as CBFV (cm s$^{-1}$) in the left and right hemispheres.

The fTCD data were analysed using a custom script in R Studio (RStudio Team, 2015). The script can be found on OSF (https://osf.io/wku3s/). The CBFV data were first down-sampled from 100 to 25 Hz by taking every fourth data-point. The data were segmented into epochs of 33 s, beginning 7 s before the presentation of the 'CLEAR' stimulus at the start of the trial (−7 s peri-stimulus time). Spiking or dropout data-points were identified as being outside of the 0.0001–0.9999 quantiles of the CBFV data. If only a single artefact data-point was identified within an epoch, it was replaced with the mean for that epoch. If more than one data-point was identified, the epoch was rejected. The CBFV was then normalized (by dividing by the mean and multiplying by 100) such that the values for CBFV become independent to the angle of insonation and the diameter of the MCA. Heart cycle integration was used to normalize the data relative to rhythmic modulations in CBFV. *Each epoch was baseline corrected using the interval from −5 to 2 s peri-stimulus time*. Finally, artefacts were identified as values below 60% and above 140% of the mean normalized CBFV—any epochs containing such artefacts were rejected.

If a participant in one session had fewer than 12 acceptable epochs for any task (i.e. more than three of the 15 epochs were rejected), the data for that task were excluded. If a participant had more than one task excluded, all data for that participant were excluded.

The CBFV from left and right sensors was averaged over all epochs at each time-point, and the mean difference (left minus right) within the period of interest was taken as the laterality index (LI). The period of interest for tasks B, C, E and F was from 6 to 23 s peri-stimulus time. For tasks A and D, the period of interest ended at 17 s to avoid activity related to overt speech production following the 'REPORT' prompt.

The LI value at each trial was also recorded, and used to calculate a standard error, which indicated how variable the lateralization was over trials. Outlier standard error values were identified using Hoaglin and Iglewicz's procedure [39]. The standard error values for every LI measurement (across all subjects, tasks and sessions; 360 values in total) were concatenated. The difference between the first

and third quartiles of the data was calculated (Q3 – Q1). In this dataset, outliers were defined as having standard error value more than 2.2 times this difference above the third quartile (Q3); e.g. the threshold limit = Q3 + 2.2 × (Q3 – Q1). Hence, if the LI value showed exceptionally high variability across trials, it was deemed to be unreliable and therefore omitted from the final analysis.

## 2.8. Departures from pre-registered methods

1. *Baseline interval*. The baseline interval was 2 s longer than that planned in the pre-registered protocol (−5 to 0 s), i.e. extending into the 'Clear mind' period. As shown in the electronic supplementary materials, this baseline gives more stable estimates of LI than the original interval.

2. *Definition of laterality index*. In our pre-registered protocol, we planned to use a peak-based method of measuring the laterality index (LI) developed by Deppe *et al.* [40], which has been standard in fTCD studies of cerebral lateralization. This involves finding the absolute peak in the difference wave within the period of interest and averaging the value of the difference over a 2 s time window centred on this peak. The major limitation of this approach is that it creates a non-normal distribution of LI values, which contributed to poor model fit in our SEM analyses, which assume normality. The mean-based method that we report here gives LI values that are highly correlated with the traditional peak-based LI (Spearman $r = 0.97$), but with a normal distribution (see electronic supplementary material for further details).

3. *Outlier detection*. In our pre-registered document, there was an error in our description of this process; we mistakenly stated we would remove outliers based on LI scores, rather than the standard error of the LI scores. Removing LI outliers would not be sensible in the context of this study, where the focus is on individual differences; it would, for instance, lead us to exclude those with atypical right-sided language laterality, who are of particular interest for our hypothesis. Our goal in outlier removal was to exclude participants with noisy data, and the LI standard error is the appropriate measure to use to achieve this goal.

4. *SEM modelling*. In addition to testing the models specified in the pre-registration document, we also tested model fit of the best-fitting model using a leave-one-out procedure, which allowed us to check whether the parameter estimates were unduly influenced by specific data-points. As described in the electronic supplementary materials, our decision to test further right-handers was prompted by discovering that there was undue influence from one left-hander, with the factor solution changing when her data were omitted. Accordingly, we present here additional analyses with 30 right-handers only, and with the full sample of 37 participants. We also computed the factor scores from the final model and plotted these to aid interpretation of the factor structure. The SEM bifactor model requires one variable to have fixed paths of 1 and 0, respectively, to the two factors. The fit of the model does not depend on which measure is used for this purpose, but the specific path estimates will vary. Given that the List Generation task was the only task with poor test–retest reliability, we present here results using Sentence Generation for the fixed paths. This follows recommendations that the strongest indicator for a specific factor should be used for the fixed paths [41].

## 2.9. Structural equation modelling

SEM, as implemented in OpenMx (https://openmx.ssri.psu.edu/), was used to test our hypotheses. All analyses were conducted in R [42]. The script used to perform this analysis can be found on OSF (https://osf.io/q8zka/). We distinguish between two sets of hypotheses: models of task effects, which concerned predictions about means, and models of person effects, which concerned covariances. As noted above, these are independent from one another. The models used to test each hypothesis are described below and can be seen in figure 4.

We will briefly describe this approach, as it not widely used in laterality research. The aim is to test how well a pre-specified model fits an observed dataset. Typically SEM is used to model covariances, but it can also be used with means. Boxes denote observed variables, two-headed arrows show variances and covariances. A triangular symbol denotes a mean value, typically set to one, with the path from the box to the triangle corresponding to the mean value for that variable. Means can be set to be equivalent by giving their paths the same label. We use capital letters for paths to means. For instance, in the Population Bias model (figure 4), all paths to the mean are set to be the same, whereas, in the Task Effect model (figure 4), the means differ from task to task, but within a task are the same from test session 1 to test session 2.

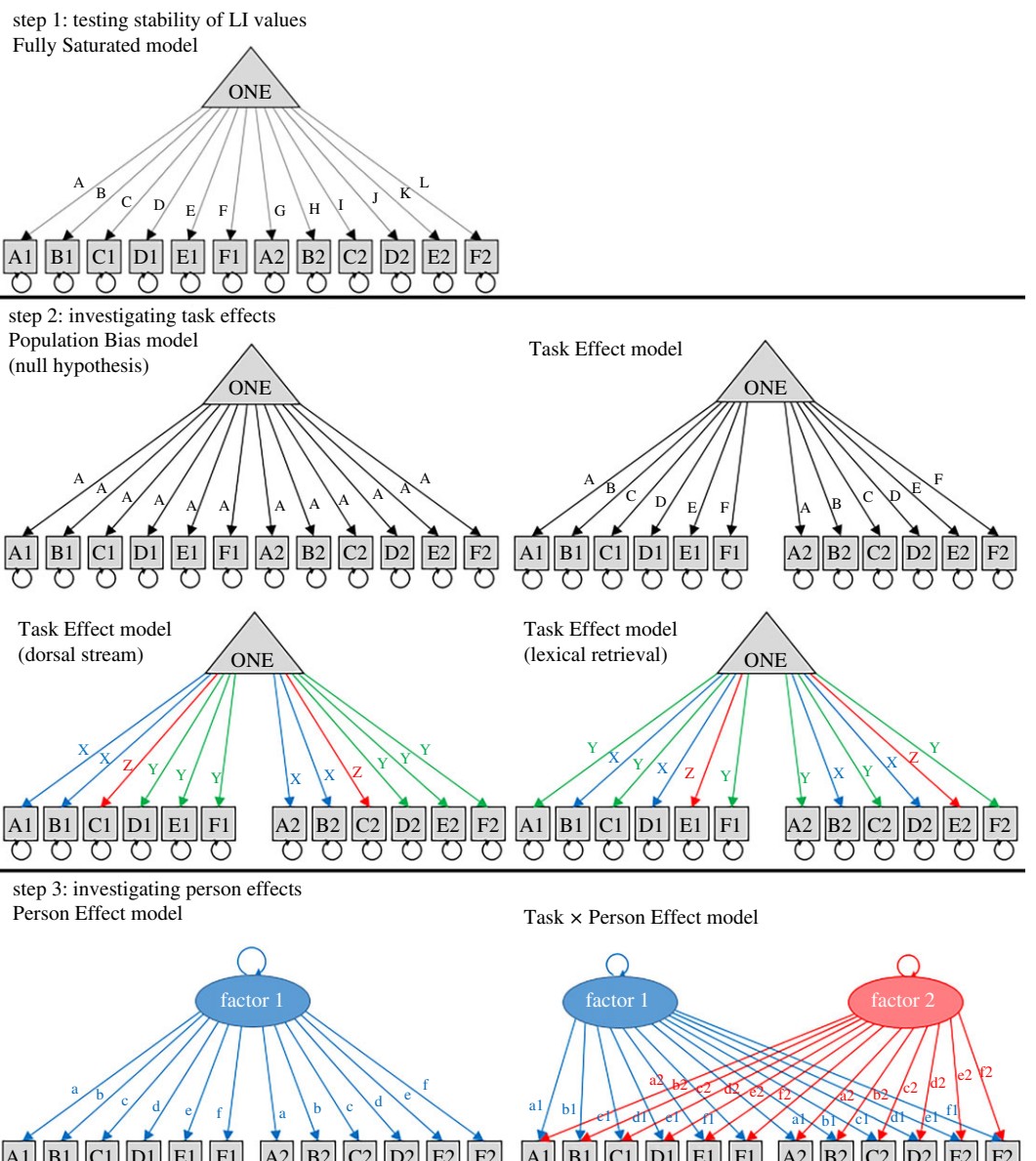

**Figure 4.** Step 1 (top): simple model of means and variances. In the 'Fully Saturated' model the means for all tasks could vary independently (tasks A – F, tested at sessions 1 and 2). This was compared to the 'Task Effect' model, where the means for each task were fixed to be the same for each session. The triangle symbol denotes that this is a model of means: covariances between values are not included in the model. Step 2 (middle): to test hypotheses relating to the LI means, the 'Population Bias' model (with means for all tasks set to be the same) was compared to the 'Task Effect' model (where means varied by task). Furthermore, to test the 'Dorsal Stream' hypothesis, a model with means for subsets of dorsal (A,B), ventral (C) and mixed tasks (D – F) were fixed (labelled as X, Z and Y). For the 'Lexical Retrieval' hypothesis, a model with means for subsets of tasks with lexical retrieval (B,D) and tasks without (A,C,F) were fixed (labelled as X and Y, respectively). Step 3 (bottom): the oval symbol denotes a common factor that determines the covariance between observed variables. To test the hypothesis relating to LI covariances, a single factor 'Person Effect' model, was compared to a two factor 'Task × Person Effect' model. To achieve model identification, one of the paths from Factor 1 to a task had to be fixed to 1, and the path from Factor 2 to that task was fixed to zero. *In our pre-registration, this fixed path was planned to be task A, but due to the low reliability of that task, it was changed in the final analysis to be task D.* The covariance between Factor 1 and Factor 2 was also set to zero. Note that the means were also modelled as shown in the Task Effect model, but this was omitted from the model diagrams here for simplicity.

An oval symbol corresponds to a latent variable linking two observed variables: covariance between two observed variables is computed as the sum of the product of the paths to those variables that are linked by an oval. Paths to latent variables are shown as lower case letters. The difference between modelling of means and covariances can be appreciated by comparing the Task Effect model and the

Person Effect model in figure 4. These look similar, but the former depicts the situation where the means for a task are constant across sessions, but covariances are not considered. Thus even if means are stable, tasks may be unreliable in the sense that individual differences are just due to noise, and the rank order of LIs of individuals is unstable. By contrast, the Person Effect model takes into account covariances, and is a test of the reliability of the measures, assessing how far individuals are consistent in their LI across occasions.

We report goodness of fit for each model relative to a 'saturated' model where all variables are unconstrained, using the comparative fit index (CFI): a high CFI indicates good model fit, and it is generally recommended that CFI needs to exceed 0.95 for the model to be regarded as a good fit to the data. We also report the root mean square error of approximation (RMSEA), which is a measure of badness of fit, and should ideally be below 0.08 [43].

Comparison of model fit to determine the most appropriate model is achieved using likelihood ratio testing. Such comparisons are valid when we have nested models. For each hypothesis, we compare two nested models computing the difference in $-2$ log likelihoods, and evaluated in terms of the difference in degrees of freedom between the two models. The difference in log likelihoods follow a $\chi^2$-distribution, so a $\chi^2$-test can be used to evaluate whether there is a statistical difference between the models. If a significant difference is found, then one model will be a better fit to the data.

In general, when comparing a model against another more complex model, good model fit corresponds to a non-significant $p$-value, which indicates that the more parsimonious model fits as well as the more complex model, despite fewer degrees of freedom. Models that estimate many parameters (and so have fewer degrees of freedom) will tend to fit the data better, and so the relative fit of models is considered using indices that take this into account. Several indices that penalize the likelihood ratio test are available, for example, Akaike's information criterion or Bayesian information criterion (BIC). Both these indices provide a value for each nested model and the lowest value among all the models is the preferred model.

### 2.9.1. Step 1: testing stability of LI values

We began with a Fully Saturated model that modelled means and variances as totally independent, as shown in figure 4 (top left). No correlations between LI values were modelled at this stage: the triangular symbol denotes that the paths reflect the mean for each observed variable. As an initial sanity check, we computed a Task Effect model where the LI value means and variances for each task (A–F) were fixed to be the same at each testing session (i.e. the means and variances for A1 = A2, B1 = B2, etc.). We predicted that the latter model would not deteriorate compared to the Fully Saturated model, indicating that we would not need to specify separate means for different test occasions.

### 2.9.2. Step 2: testing models of means

Our first hypothesis proposed that a significant task effect on LI value would be observed; i.e. that the mean LI values would vary between the six different tasks (tasks A–F). This was assessed by comparing the two models shown in row 2 of figure 4: the Population Bias model and the Task Effect model.

The Task Effect model was then used as a baseline comparison model to test two more specific sub-hypotheses regarding which tasks would show the strongest lateralization. In each case, we divided tasks into three subsets, and fixed the means and variances for the tasks within each subset to be the same. We adopted this approach to test the Dorsal Stream hypothesis and the Lexical Retrieval hypothesis.

### 2.9.3. Step 3: testing models of covariances

Two models of covariance were compared (figure 4, bottom). First, a Person Effect model was computed where covariance was predicted by a single factor, i.e. was similar across all language tasks. This was compared with a Person by Task Effect model, with two covariance factors. The Person Effect (single factor) model is nested within the Task × Person Effect (bifactor) model, and so their relative fit can be assessed by subtraction of negative log likelihoods.

## 3. Results

All data are available on OSF (https://osf.io/s9kx6/). Results from the pre-registered analysis protocol (i.e. using the first 30 participants only) are shown in the electronic supplementary materials. As noted

**Table 1.** Behavioural data for tasks B, C, E and F. The table shows mean percentage accuracy and reaction times (with s.d.), and results of t-tests comparing session 1 with session 2 for each measure. The number of omitted responses is reported as a percentage of all events. B, Phonological Decision; C, Semantic Decision; E, Sentence Comprehension; F, Syntactic Decision.

| measure | session | task B | task C | task E | task F |
|---|---|---|---|---|---|
| accuracy (%) | 1 | 91.3 (5.55) | 95.9 (3.08) | 92.5 (4.81) | 89.6 (8.31) |
| | 2 | 93.3 (4.28) | 95.0 (3.06) | 94.2 (3.79) | 89.4 (8.28) |
| | 1 versus 2 | $t = -3.27$, $p = 0.002$ | $t = 1.61$, $p = 0.115$ | $t = -2.70$, $p = 0.011$ | $t = -0.07$, $p = 0.944$ |
| reaction times (s) | 1 | 1.66 (0.22) | 1.14 (0.2) | 2.17 (0.12) | 0.334 (0.08) |
| | 2 | 1.49 (0.21) | 1.06 (0.2) | 2.11 (0.15) | 0.329 (0.07) |
| | 1 versus 2 | $t = 8.73$, $p < 0.001$ | $t = 4.77$, $p < 0.001$ | $t = 3.27$, $p = 0.002$ | $t = 0.64$, $p = 0.528$ |
| omitted responses (%) | 1 | 2.34 | 0.84 | 2.79 | 4.20 |
| | 2 | 0.78 | 0.60 | 1.62 | 4.44 |

above, the factor solution from this sample was unstable and unduly influenced by one left-hander. We report here the results based on the final sample of 30 right-handers and seven left-handers, which gives a stable solution, and we include exploratory analyses relating the findings to handedness. The LI values reported here are based on the mean difference between left and right CBFV, as this gives normally distributed variables, but the results are highly similar when the non-normal peak-based LIs are used instead. The analysis script provided on OSF (https://osf.io/q8zka/) facilitates comparisons between different analytic pathways.

## 3.1. Behavioural results

We did not have specific predictions for the behavioural results, but present them here for completeness. For List Generation (A) and Sentence Generation (D), the number of words spoken per trial was recorded. The number of words spoken in both tasks and sessions were very similar: for task A, session 1, mean = 9.5, s.d. = 0.42, session 2, mean = 9.6, s.d. = 0.29; for task D, session 1, mean = 9.2, s.d. = 1.21, session 2, mean = 9.4, s.d. = 1.24. A repeated measures ANOVA showed no significant effects of task ($F_{1,36} = 1.22$, $p = 0.278$) on the number of words spoken, but there was a significant effect of session ($F_{1,36} = 5.73$, $p = 0.022$) as participants produced more words at the second session. Trials where participants failed to respond, or responded too early were excluded from analysis: these constituted less than 0.1% of trials.

For decision-making tasks (B, C, E and F), the accuracy and RT of each response, and the number of omitted responses, were recorded (table 1). Note that for task F participants were required to wait until the end of the word sequence before responding, and had only a second to respond; this accounts for the fast reaction times and relatively high number of omitted responses in task F.

The Phonological Decision and Sentence Comprehension tasks (tasks B and E) showed evidence of practice effects, as both accuracy and reaction times improved, and the number of omitted responses fell from session 1 to session 2.

## 3.2. Lateralization results

Five outlier LI values were excluded where the standard error across trials was above the upper cut-off. Six LI values were excluded because a subject had less than 12 useable trials for a given task in a given session. The remaining data for these participants were retained in the analysis. Excluded data-points are shown as red dots in figure 5.

Figure 5 shows the distribution of LIs as a pirate plot [44]. Task D (Sentence Generation) showed the strongest left lateralization. Shapiro–Wilks normality tests showed that LI values for all 12 conditions were normally distributed. One sample t-tests (testing for mean greater than 0) showed that all conditions were significantly left lateralized, except task F (Syntactic Decision; session 1: $t_{33} = 0.77$, $p = 0.224$; session 2: $t_{36} = 0.33$, $p = 0.373$).

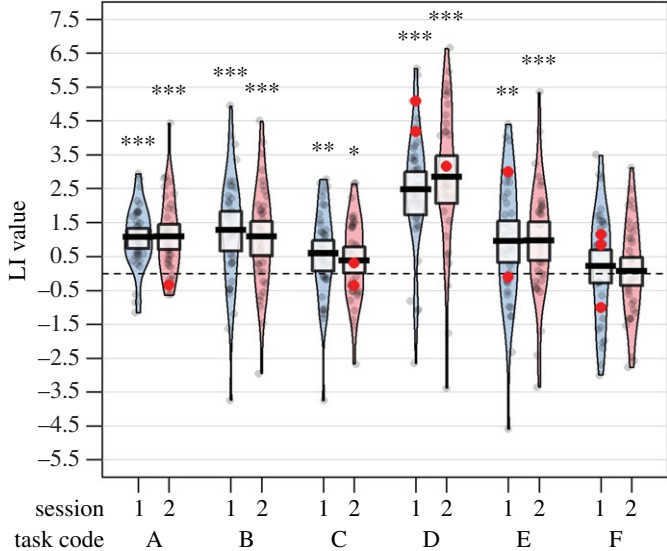

**Figure 5.** Pirate plot of LI values for all tasks (A–F) and sessions (blue = session 1, pink = session 2). Excluded data-points are shown in red. Asterisks show results of Wilcoxon tests comparing the LI values of the group (omitting excluded data-points) to zero ($*p < 0.05$; $**p < 0.01$; $***p < 0.001$).

Figure 6 shows a correlation matrix of LI values for all tasks and sessions. Test–retest correlations varied between tasks. Task A (List Generation) had poor test–retest reliability (Pearson's $r = 0.13$), and low correlations with other tasks. Test–retest reliability for other tasks ranged from $r = 0.54$ to $0.84$. Tasks B, C, D and E were strongly intercorrelated. Task F (Syntactic Decision) had high test–retest reliability ($r = 0.76$) but relatively low correlations with other tasks.

## 3.3. Structural equation modelling

The LI data were entered into the SEM analysis to test hypotheses about the group mean LI values and covariances in LI values across subjects. Table 2 summarizes the SEM results.

### 3.3.1. Step 1: testing stability of LI values

As shown in table 2, the fit of all the means-only models was very poor. This is to be expected, as these models ignore covariances, and, as indicated in figure 6, there are substantial correlations both between and within tasks. Our interest at this point, however, is in the relative fit of different models of means, rather than overall model fit. The Fully Saturated model (with free means and variances) was compared to the Task Effect model, which fixed the means and variances for each task to be stable over sessions (i.e. A1 = A2, B1 = B2, etc.). The Task Effect model fit did not deteriorate significantly from that of the Fully Saturated model, supporting the hypothesis that LI means for each task were stable across sessions.

### 3.3.2. Step 2: testing models of means

To demonstrate whether LI means differed between tasks, the Task Effect model (with different means for each task) was compared to the Population Bias model (with means fixed to be the same for all tasks). This may be seen as a null hypothesis that treats all tasks as equivalent measures of laterality. The Population Bias model gave significantly worse fit (table 2), supporting the hypothesis that LI means differed between tasks.

Two further models were compared to the Task Effect model. The Dorsal Stream model categorized the language tasks according to the involvement of the dorsal or ventral stream. Tasks A and B were categorized as involving strong dorsal stream activity, task C as strong ventral stream activity, and tasks D, E and F as intermediate (hence, means for AB > DEF > C). This model gave significantly poorer fit than the Task Effect model—as is evident from figure 5, which shows relatively weak lateralization for tasks A and B compared to task D. The Lexical Retrieval model did not fare any better. This categorized tasks B and D as involving strong lexical retrieval, whereas tasks A, C and F

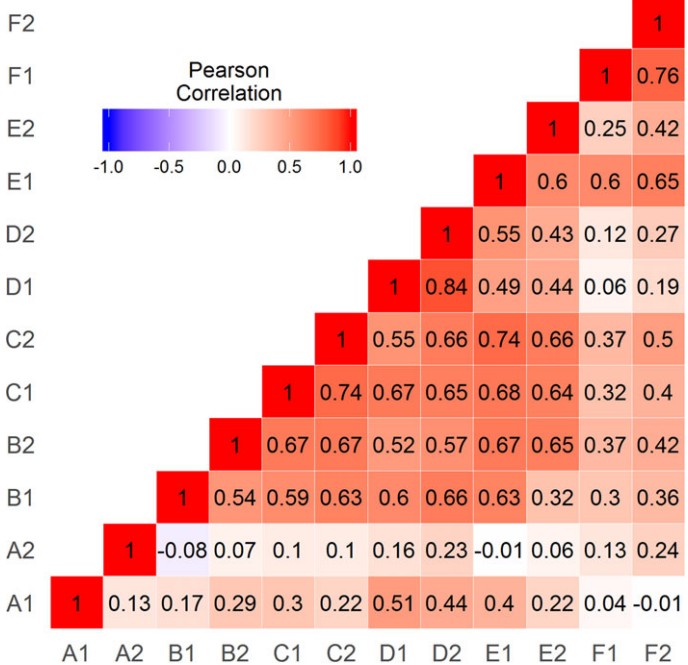

**Figure 6.** Correlation matrix for LIs from the six language tasks given on two occasions.

did not involve lexical retrieval, and task E was difficult to classify and so was considered as independent of the other measures (BD > ACF). Again, this model gave a worse fit than the Task Effect model, indicating that, while laterality varied between tasks, it did not fit either of the predicted patterns. Note, however, that the pre-registered tests specified for both theories have some limitations, as discussed further below.

### 3.3.3. Step 3: testing models of covariances

At Step 3, we tested whether the covariances between tasks had a single factor structure (Person Effect model) or a bifactor structure (Task by Person Effect model). Not surprisingly, given the strong correlations in figure 6, both within and across tasks, the Person Effect model gave a substantially better fit than the Task Effect model (table 2); nevertheless, the overall fit of this model was poor. The Task by Person Effect model gave a significantly improved fit. A plot of the two factors is shown in figure 7: note that, although the model fit is not affected by task selection, the factor scores depend on which task has fixed paths to the factors. The paths for the case when Sentence Generation is fixed are shown in table 3. It can be seen that List Generation has only a weak loading on Factor 1, whereas Phonological Decision, Semantic Decision and Sentence Comprehension have moderate loadings on both factors. Syntactic Decision has a strong loading on Factor 2 but does not load on Factor 1, reflecting the weak correlation of this task with Sentence Generation.

In our original analysis with 30 participants, a similar factor structure was observed, but there was a concern that this depended solely on a single left-handed participant (see the electronic supplementary material). With the larger sample of 37 participants, the bifactor (Task by Person Effect) model was superior in all runs of a leave-one-out analysis. The bifactor model was also the best-fitting model when only the 30 right-handers were included in the analysis. Nevertheless, it is clear from figure 7 that the two factors were highly intercorrelated, and the impression is that the bifactor solution is heavily affected by some influential cases. Cook's distance identified four bivariate outliers, marked with circles in figure 7: all four outliers were left-handers. When the analysis was re-run omitting these cases, the single factor model gave a better model fit when all $N = 33$ subjects were included (single factor BIC = $-142.7$, bifactor BIC = $-138.6$), and in all but one run of the leave-one-out analysis.

We can conclude from this analysis that, although univariate normality was satisfactory, our data did not meet conditions of multivariate normality; this leads to the conclusion that the sample is not homogeneous, but contains a mixture of laterality patterns. We discuss the implications of this finding below.

**Table 2.** Model fit statistics from structural equation models and model comparisons. −2logL, −2 log likelihoods; d.f., degrees of freedom; BIC, Bayesian information criterion; CFI, comparative fit index; RMSEA, root mean square error of approximation.

| model | description | −2logL | d.f. | BIC | CFI | RMSEA | $\chi^2$-test compared to | p |
|---|---|---|---|---|---|---|---|---|
| Fully Saturated model | free means and variances | 1568.8 | 409 | 92.0 | NA | NA | — | NA |
| Task Effect model | stable means and variances | 1575.2 | 421 | 55.0 | 0.023 | 0.289 | Fully Saturated model | 0.896 |
| Population Bias model | equal means and variances | 1706.5 | 431 | 150.2 | 0 | 0.334 | Task Effect model | <0.001 |
| Dorsal Stream model | means for tasks AB > DEF > C | 1657.2 | 427 | 115.4 | 0 | 0.320 | Task Effect model | <0.001 |
| Lexical Retrieval model | means for tasks BD > ACF | 1630.2 | 427 | 88.4 | 0 | 0.306 | Task Effect model | <0.001 |
| Person Effect model | covariances have one factor structure | 1378.0 | 415 | −120.6 | 0.796 | 0.137 | Task Effect model | <0.001 |
| Task × Person Effect model | covariances have bifactor structure | 1337.7 | 410 | −142.8 | 0.939 | 0.078 | Person Effect model | <0.001 |

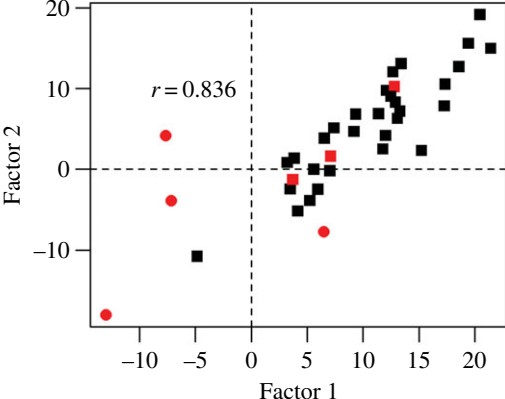

**Figure 7.** Correlation between two factors from the bifactor (Task by Person Effect) model, with left-handers shown in red, and bivariate outliers as circles.

**Table 3.** Path weightings (and 95% confidence intervals) from each latent factor (Factor 1 and Factor 2) to each task (A to F) from the winning bifactor model.

| task | Factor 1 | | Factor 2 | |
| --- | --- | --- | --- | --- |
| | path | 95% CI | path | 95% CI |
| A. List Generation | 0.18 | 0.05 to 0.31 | −0.01 | −0.27 to 0.24 |
| B. Phonological Decision | 0.60 | 0.39 to 0.80 | 0.53 | 0.19 to 0.88 |
| C. Semantic Decision | 0.53 | 0.36 to 0.70 | 0.53 | 0.24 to 0.82 |
| D. Sentence Generation | 1.00 | Fixed | 0.00 | Fixed |
| E. Sentence Comprehension | 0.56 | 0.30 to 0.83 | 0.95 | 0.53 to 1.36 |
| F. Syntactic Decision | 0.13 | −0.13 to 0.40 | 1.16 | 0.76 to 1.56 |

In a final exploratory analysis, we tested whether the poor reliability of the List Generation task (A) may have affected the SEM results. Re-running the SEM analyses with five tasks only (excluding List Generation) did not change the rank order of the SEM model fits, or the significance of the model comparisons. The bifactor model remained the best fit to the data, with path strengths very similar to those shown in table 3. The results of this analysis are shown in the electronic supplementary material.

## 4. Discussion

The question of whether cerebral lateralization is a unitary function may be interpreted at two levels: at the population level, we may ask whether all language tasks show a similar degree of lateralization, and at the individual level, whether people show consistent differences in laterality profiles across tasks.

Although we used formal modelling to address these questions, a good insight into the answers can be obtained by viewing figures 5 and 6. Figure 5 shows clear differences from task to task in the strength of cerebral lateralization within the MCA territory, whereas figure 6 shows moderate-to-good test–retest reliability for all but one task, coupled with significant cross-task correlations. This pattern of correlations is indicative of a Person by Task interaction (as hypothesized in figure 1): there is a cluster of tasks (B, C, D and E) that have strong cross-task correlations, indicating that they are driven by a common lateralizing factor. By contrast, Task F has weak correlations with these tasks, despite having high test–retest reliability. This suggests that lateralization on this task is driven by a second, independent factor.

The SEM analyses provided an economical approach for considering a range of hypotheses within a single framework, allowing us to test statistically whether this pattern between tasks was significant. SEM was used as it allowed us to formalize our hypotheses in statistical models that could then be compared using likelihood ratio significance tests, rather than making subjective inferences from the pattern of means and correlations alone. Regarding means, as expected, a null hypothesis of no difference between tasks could be convincingly rejected. However, the specific patterns that we

predicted should be seen on the basis of two existing models—the Dorsal Stream model and the Lexical Retrieval model—did not give a good fit. It could be argued that the data are, in fact, consistent with the Dorsal Stream model, insofar as the three tasks that involved implicit or explicit generation of speech— List Generation, Phonological Decision and Sentence Generation—were the ones that showed the strongest lateralization (figure 5). The poor fit of the Dorsal Stream model was, in part, due to the fact that Sentence Generation was judged to implicate both streams, and was not, therefore, predicted to be as strongly lateralized as tasks with weaker semantic demands. However, it clearly makes demands on the phonological-articulatory system, and with hindsight it could be argued that in terms of articulatory complexity it was more demanding than the other tasks. A key question is whether blood flow measured using fTCD reflects the average of activity in a lateralized dorsal stream and a bilateral ventral stream, or whether the absolute dorsal stream activity is the main factor affecting the LI. In future, we plan studies to address this question using fMRI.

More generally, based on the pattern of results observed in this study, it appears that whole-hemisphere lateralization as measured by fTCD is driven most strongly by the generation of meaningful, connected speech (e.g. Sentence Generation). Lateralization for this task was stronger than for automatic, non-propositional speech (List Generation) or implicit sub-vocalization (Phonological Decision). By contrast, lateralization was non-significant for the Syntactic Decision task.

We would, however, emphasize the need for caution in treating any one task as an indicator of a particular language function: it is evident that even minor modifications to task demands may affect laterality, particularly when sample size is relatively small. For instance, in a related study with a different sample of people, we recently found that List Generation was not lateralized [26]. In that study we interleaved a simple number generation (counting) task with trials of Sentence Generation, whereas in the current study, List Generation was administered in a separate block, with the type of list (numbers, days of the week, months of the year) varied to engage the participants' attention throughout the block. Although the counting task used by Woodhead et al. [26] was not significantly lateralized, it had good split-half reliability and was significantly correlated with Sentence Generation, whereas the List Generation task used in the current study was the only task to show poor test–retest reliability and relatively weak correlations with other tasks. Furthermore, our Semantic Decision task was designed to tap into similar semantic processes as the pyramids and palm trees test [45], but resulted in weaker LIs than seen in a study by Bruckert [32] using the pyramids and palm trees task. It could be that the two-alternative forced choice task used in that study was more demanding than our match/no-match decision, but this kind of difference cautions us about relying on a single test to indicate a type of linguistic processing.

One convincing point to emerge from the analysis of mean data is that most language tasks (B, C, D, E and F) showed stable lateralization measured in different sessions, but they differed in terms of the strength of left lateralization. The question of why task A (List Generation) had such low test–retest reliability remains open, but it is possible that the covert generation of speech sequences was not sufficiently engaging to elicit robust, reliable brain activity. The behavioural measure used for this task (number of words spoken) is not a very useful index of engagement in the task, as the overt speech generation could be performed perfectly even if the participant did not covertly generate the words at all. This is in contrast to task D (Sentence Generation) where covert generation would aid the participant's overt sentence reporting.

We acknowledge that the SEM models of means had poor fit but they were not required to fit the data well as they were probably an over-simplification of the underlying structure. We only sought to discount them as viable, more parsimonious models than the later complex models. The poor model fit limits their interpretability; but our intention was not to interpret them in isolation. Rather, we view them as stepping stones along the way, by identifying the optimal structure to explain the LI means prior to adding the covariance structure. Only the later models achieved satisfactory fit for interpretation.

We turn next to the findings concerning covariances. It has been argued that fTCD is not useful for studying cerebral lateralization because it is unreliable [22], but our data support those of Stroobant & Vingerhoets [46] in demonstrating that there is significant individual variation in language laterality between people that cannot just be attributed to noise. Furthermore, by moving from a definition of laterality based on a peak in the L-R difference wave to a definition based on mean L-R difference within a period of interest, we avoid the problem that can arise when laterality is forced into a non-normal distribution (see also [26]). As shown in figure 5 and our tests of normality, when mean L-R difference is used, the distribution of LI values is normal.

The SEM also tested whether a single factor could explain individual differences in language lateralization. At first glance, the results suggested that this was not the case: the bifactor (Task by

Person Effect) model showed superior fit over a single factor (Person Effect) model. This was the conclusion suggested by our initial pre-registered analysis, based just on a sample of 30 individuals. A leave-one-out analysis, however, made us cautious about accepting that result at face value, because the factor structure changed when a single left-hander with strongly complementary laterality on two tasks was excluded. For this reason, we collected more data, adding seven right-handers to the sample. With this larger sample, we again found superiority for a bifactor solution, regardless of whether we included only right-handers or the full sample including left-handers. Yet there remained misgivings about the generalizability of the result, not least because the two factors were highly correlated (Pearson's $r = 0.84$). A scatterplot of the two factors revealed a number of bivariate outliers and, as with our initial analysis, the pattern of results relied on which participants were included. Of course, it is not surprising that removing participants with the strongest dissociation between factors changes the factor structure: the point we wish to make is not that the results can alter in this way, but rather that the pattern of our SEM findings appears driven by heterogeneity within the sample, reflected in the presence of bivariate outliers.

These results indicate that SEM can be useful for studying individual variation in patterns of laterality, but it needs to be interpreted with caution. It can be tempting to focus on achieving a good model fit, but, as Hooper *et al.* [47] noted, the purpose of SEM should be model-testing, and even a well-fitting model may be suboptimal. Our data illustrate that point clearly: having found support for a bifactor model with no paths between the latent factors, we expected that we would have two independent factors that were binormally distributed. By performing further checks in terms of a leave-one-out analysis and visualization of the two extracted factors, it became clear that this was not the case, and that our data contained a mixture of two subgroups. One the one hand, we can conclude that SEM is not optimal for investigating multivariate aspects of laterality, because it assumes multivariate normality. On the other hand, by applying SEM and studying the anomalies that resulted, we gained insights into the heterogeneous nature of our sample.

The answer to the question of whether laterality is a unitary function is that, clearly, there are some individuals in whom laterality is different for different aspects of language. It is not, however, the case that there are two factors that act independently in the general population. Rather, the majority of people appear to have language laterality driven by a single process affecting all types of task, with a minority showing fractionation of language asymmetry. This is consistent with observations of discrepant hemispheric dominance for different language functions [3–5], which occurs in a small minority of participants.

The pattern of results is also consistent with accounts of laterality that postulate qualitative rather than just quantitative differences between individuals. Theoretical accounts have mostly focused on a single dimension, arguing for laterality subgroups on the basis of non-normal distributions of scores (e.g. [8]). Our results suggest that atypical laterality may be easier to identify when more than one language measure is considered, as detection of bivariate outliers can be effective with smaller samples than those required for detecting mixtures of distributions.

An association between atypical laterality and left-handedness has been established for many years, ever because early observations were made of superior recovery from aphasia after gun-shot wounds in left-handers [48]. However, most of the emphasis has been on atypical laterality in the sense of having language mediated by the right hemisphere. Although the number of left-handers in our sample is too small for numeric analysis, the fact that all of the four bivariate outliers were left-handers is a striking departure from chance and compatible with the idea that language lateralization is more likely to be multifactorial in left-handers than right-handers.

Further studies are needed to establish the key characteristics of tasks that index the two factors seen in some people, but we offer here some speculations. The main contributor to the second factor was the Syntactic Decision task, which differed from the other tasks in several regards. It used unfamiliar, non-word stimuli, and required the listener to identify syntactic errors. It was one of two receptive language tasks that involved processing of auditory language: the other was sentence comprehension, which had moderately strong loadings on the second factor. Perhaps the most surprising finding from this study is the fact that the one task that loaded on to the second factor (Syntactic Decision) was not lateralized, yet showed high test–retest reliability ($R = 0.76$). We had anticipated that a lack of lateralization on a task might be a consequence of noisy data giving poor test reliability—or alternatively a lack of individual variation if both hemispheres contributed equally in most people. Our data suggest that individuals do vary in the hemisphere used when doing the syntactic judgement task, and that this bias is reliable, but that it is not systematic across the population. A similar finding was recently reported in an fMRI study investigating the validity and reliability of different language paradigms, which

showed that a picture naming task showed high test–retest reliability despite not being lateralized [49]. These findings demonstrate that strength as well as direction of lateralization for a task are both stable traits.

### 4.1. Limitations

As noted above, the principal limitation of fTCD is that it does not allow one to localize lateralized activity within a hemisphere. LI as measured in fTCD reflects only the difference in blood flow within the left and right MCA territories, and is insensitive to activation outside of those areas. In future work, we plan to extend this line of investigation to consider whether similar patterns of lateralization can be seen using comparable tasks with fMRI. The benefit of fTCD is that it is relatively inexpensive and quick to administer, and so enables us to gather data that can be used as a basis for developing a more hypothesis-driven approach that can then be extended and validated with fMRI.

A further limitation is that we lacked statistical power or range of measures that would be needed to evaluate more complex models. The bifactor model that gave the best fit in our study must be interpreted with caution. It will need to be replicated in larger samples and shown to generalize to new tasks—it remains a possibility that using a different set of tasks would reveal different or further fractionation of language lateralization. Furthermore, although we have shown a bifactor model is a better fit than a single factor model, it is possible that more than two factors are needed to explain the full range of patterns of language lateralization.

### 4.2. Summary

In summary, these results indicate that there are meaningful differences in language lateralization between tasks, and meaningful individual variability in lateralization that is not simply due to measurement error. Even when a language-related task is not left lateralized, there are stable individual differences in the contribution of the two hemispheres. SEM of individual variability indicated that although a two-factor model gave a better fit than a single factor model, the effect was driven by a small subset of participants with discrepant laterality, and a single factor could account for variation in the majority of participants. Overall, our findings suggest there are qualitative as well as quantitative differences between people in laterality across tasks, and that consideration of asymmetry profiles on several tasks together can help identify cases of atypical laterality.

Ethics. All participants gave written informed consent. Procedures were approved by the University of Oxford's Medical Sciences Interdivisional Research Ethics Committee (approval no. R40410/RE004).

Data accessibility. All task materials, analysis scripts and anonymized data are available on Open Science Framework (https://osf.io/tkpm2/, doi:10.17605/OSF.IO/TKPM2).

Authors' contributions. Z.W. was involved in conceptualization, investigation, formal analysis, data curation and writing the manuscript. A.B. and A.W. were involved in conceptualization, production of materials, investigation and manuscript editing. P.T. was involved in data analysis, visualization and manuscript editing. D.B. was involved in conceptualization, formal analysis, manuscript editing, supervision and funding acquisition. All authors gave final approval for publication.

Competing interests. We declare we have no competing interests.

Funding. This study was supported by the Wellcome Trust (082498) and by an Advanced Grant from the European Research Council (694189).

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
