## [Reviewer comments · Royal Society Open Science]

Review History

RSOS-181801.R0 (Original submission)

Review form: Reviewer 1 (Chris Tailby)

Is the manuscript scientifically sound in its present form?

Yes

Are the interpretations and conclusions justified by the results?

Yes

Is the language acceptable?

Yes

Is it clear how to access all supporting data?

Yes

Do you have any ethical concerns with this paper?

No

Have you any concerns about statistical analyses in this paper?

No

Recommendation?

Accept with minor revision (please list in comments)

Comments to the Author(s)

This manuscript uses a battery of 6 different language tasks administered on two occasions to examine patterns of hemispheric lateralisation via fTCD. Structural equation modelling is used to tease apart different potential accounts of the observed data. The main conclusion is that there are a small subset of individuals in whom lateralisation on a given task is not explained by a single laterality factor, implying that there are important individual differences in task-dependent lateralisation.

This is a very clearly written article, cleanly laid out and argued with good use of tables of figures.

The task main effect hypothesis (i.e. not all tasks are created equal from a laterality perspective) is a bit of a 'straw man', but I see the utility in the larger modelling scheme. I like the design and modelling approach used to address the question of task dependent individual differences (manifest in the comparison of the Person Effect Model versus the Task by Person Effect Model).

For Task F, I assume the button press was used to indicate whether participants thought the 'jabberwocky' was a sentence or not? This is not clear from the text.

In Figure 1, I don't understand why the data clouds in the right-hand column aren't centred on the cross-hairs defined by the red dashed lines (which I understand to show task means in the models). Shouldn't they be?

The justifications for the departures (e.g. sample, data handling) from the pre-registered approach are valid.

It is reassuring to see that LIs across repetitions of a given task are (at least relatively) stable.

Regarding the question of multivariate outliers (Fig 7), do the authors know/can they show that repeatability across time was comparable for the 'outlier' cases as for the non-outlier cases. In other words, are the outliers 'stably distinctive' in their profile, or is part of their distinctiveness their variability over time? Knowing that they are 'stably distinctive' would reinforce the take home message about individual differences.

In discussion: "A key question is whether blood flow measured using fTCD reflects the average of activity in a lateralised dorsal stream and a bilateral ventral stream, or whether the absolute dorsal stream activity is the main factor affecting the LI. In future we plan studies to address this question using fMRI." This is a good question; indeed, I wondered (I am just making an observation here) whether fTCD was the best tool to answer the questions posed in this study (as opposed to say fMRI).

I agree with the statement: "The answer to the question of whether laterality is a unitary function is that, clearly, there are some individuals in whom laterality is different for different aspects of language. It is not, however, the case that there are two factors that act independently in the general population. Rather, the majority of people appear to have language laterality driven by a single process affecting all types of task, with a minority showing fractionation of language asymmetry." This paragraph could do with some citations as this (individual cases who do not

conform to the normal, uniformly left dominant picture) has been noted before.

Review form: Reviewer 2 (Mohamed Seghier)

Is the manuscript scientifically sound in its present form?

Yes

Are the interpretations and conclusions justified by the results?

No

Is the language acceptable?

Yes

Is it clear how to access all supporting data?

Yes

Do you have any ethical concerns with this paper?

No

Have you any concerns about statistical analyses in this paper?

Yes

Recommendation?

Major revision is needed (please make suggestions in comments)

Comments to the Author(s)

Reviewed by Mohamed Seghier:

In this manuscript, Woodhead and colleagues used SEM to model between-task and between-subject variability in fTCD-based laterality indices. They showed that variability in laterality scores is meaningful and can be explained by task-by-subject factors.

The topic of this paper is very interesting; it is a nice addition to the current literature (debate?) about modelling/explaining the between-subject variability in lateralization. I have some minor comments about the current version of this manuscript and I hope the authors will find them useful.

As authors know, in addition to task and subjects factors, laterality also varies with brain region. However this spatial dimension is lost when using fTCD. So even if laterality scores or indices were stable across tasks, this does not necessarily mean that the spatial patterns were similar across tasks. ROI localization has strong impact on reliability and consistency between tasks, and many previous studies have shown significant between-region associations (spatial covariance) in lateralization at the cluster level (e.g. Seghier et al. 2011 HBM; Cai et al. 2010 Cereb Cortex) as well as at the voxel level (e.g. Seghier and Price, 2013 Brain Lang).

As acknowledged by the authors in Page 31, I don't think fTCD is suitable to test/compare predictions based on theoretical accounts that differ in terms of processing pathways (Hickok and Poeppel's dual route model of speech processing versus Dhanjal et al.'s lexical retrieval account). This section about theoretical accounts is not compelling in my opinion and I recommend to the authors to move it to the supplementary material (but I leave this to their discretion): (1) there is no strong rationale why only these two accounts have been selected among other previous but competing accounts; (2) the two accounts did not fully explain/predict the possible differences in

laterality between tasks (e.g. see Task E); (3) the task where both accounts provided contrasting predictions was a task with poor test-retest reliability (Task A); (4) the RMSEA associated with both models (Table 2) was very poor which may suggest that those two models were ‘poor’ models given the fTCD data; (5) both models gave a worse fit than the Task Effect model; (6) there was heavy reliance on ‘reverse inference’ to generate predictions based on each account (e.g. which brain regions/streams were most likely implicated by a given task). Again, my point here is that this model comparison between predictions based on theoretical accounts would be an interesting question for future fMRI studies but not fTCD.

Did the authors select those six tasks from preliminary/piloting studies? It would be interesting for readers to know why you went for this set of tasks given the high level of variation and inconsistency between different tasks that you discussed in your 2017 systematic review (Peer J 2017). Based on that 2017 review, some useful recommendations were made about task design for fMRI studies and I’m wondering whether those recommendations are also valid for optimal task design when using fTCD.

In the discussion section, please add a succinct paragraph about the usefulness of modeling laterality data with SEM: e.g. highlighting the kind of inferences one can make with SEM beyond what can already be inferred from Figures 5 and 6. Also, do you recommend SEM only for homogeneous samples given the significant impact of one left-hander (outliers?) on the original findings?

Page 5: how did you generate the synthetic data of Figure 1? Are these simulations based on the output of the selected models (similar to laterality simulations of Page 10)? Moreover, it would be useful for the reader to discuss how the empirical data (Figure 6) compared with your predictions from the simulated data (Figure 1). For instance, cross-task correlations were predicted for models with a ‘Person’ effect only (Figure 1), and real data showed significant cross-task correlations (Figure 6) that were better explained by a bifactor model.

The RMSEA values in Table 2 may indicate poor fit (except for the last model, all values are well above the upper limit of 0.08; see Hooper et al. 2008 “The Electronic Journal of Business Research Methods”). This could hint to a poor fit of the data covariance matrix or a kind of redundancy in the parametrization of the models. I recommend to the authors to succinctly mention this issue in the Discussion section, given current limitations in reporting RMSEA (and its interpretation) for studies with small degrees of freedom (see Kenny et al. 2015 “Sociological Methods & Research”, 44(3):486-507).

It is puzzling why Task A (List Generation) showed poor test-retest reliability. In the behavioral analysis of Page 23, there was a significant session effect in the ANOVA analysis. Can this explain the poor reliability?

Task F (Syntactic Decision) showed weak lateralisation but still good test-retest reliability. I’m curious to know whether the absolute means (not the difference) of CBFV for left and right hemispheres were higher for Task F than Task A, which could maybe suggest higher blood flow velocity and hence robust signals during the execution of Task F.

Task F was not lateralized but showed high test-retest reliability. Recently, Wilson et al. (2017 NeuroImage: Clinical) found similar result with a different task: a picture naming task with high test-retest reproducibility despite showing no lateralization in fMRI. This can be mentioned in the Discussion section (Page 34).

In Table 1, please double check the t scores for the paired t-test comparisons between RTs. For example, Task F yielded identical means but the t score was 0.64.

Some CFI indices were negative in Table 2. CFI should range between 0.0 and 1.0.

Decision letter (RSOS-181801.R0)

15-Jan-2019

Dear Dr Woodhead,

The editors assigned to your paper ("Testing the unitary theory of language lateralisation using functional transcranial Doppler sonography in adults") have now received comments from reviewers. We would like you to revise your paper in accordance with the referee and Associate Editor suggestions which can be found below (not including confidential reports to the Editor). Please note this decision does not guarantee eventual acceptance.

Please submit a copy of your revised paper before 07-Feb-2019. Please note that the revision deadline will expire at 00.00am on this date. If we do not hear from you within this time then it will be assumed that the paper has been withdrawn. In exceptional circumstances, extensions may be possible if agreed with the Editorial Office in advance. We do not allow multiple rounds of revision so we urge you to make every effort to fully address all of the comments at this stage. If deemed necessary by the Editors, your manuscript will be sent back to one or more of the original reviewers for assessment. If the original reviewers are not available, we may invite new reviewers.

- Data accessibility

It is a condition of publication that all supporting data are made available either as supplementary information or preferably in a suitable permanent repository. The data accessibility section should state where the article's supporting data can be accessed. This section should also include details, where possible of where to access other relevant research materials

such as statistical tools, protocols, software etc can be accessed. If the data have been deposited in an external repository this section should list the database, accession number and link to the DOI for all data from the article that have been made publicly available. Data sets that have been deposited in an external repository and have a DOI should also be appropriately cited in the manuscript and included in the reference list.

If you wish to submit your supporting data or code to Dryad (<http://datadryad.org/>), or modify your current submission to dryad, please use the following link:
<http://datadryad.org/submit?journalID=RSOS&manu=RSOS-181801>

- **Competing interests**

- **Authors' contributions**

- **Acknowledgements**

- **Funding statement**

on behalf of Dr Shirley-Ann Rüschemeyer (Associate Editor) and Professor Antonia Hamilton (Subject Editor)
openscience@royalsociety.org

Associate Editor's comments (Dr Shirley-Ann Rüschemeyer):

Dear Authors,

Thank you again for considering Royal Society Open Science as an outlet for your work.

Two reviewers and I have read your manuscript, and are in agreement that the work is suitable for publication. The reviewers have both brought up a number of methodological and theoretical questions, that I would ask you to address in the final version of your manuscript. The reviewers will be invited to comment on your final manuscript.

With best wishes,

Comments to Author:

Reviewers' Comments to Author:

Reviewer: 1

Comments to the Author(s)

This manuscript uses a battery of 6 different language tasks administered on two occasions to examine patterns of hemispheric lateralisation via fTCD. Structural equation modelling is used to tease apart different potential accounts of the observed data. The main conclusion is that there are a small subset of individuals in whom lateralisation on a given task is not explained by a single laterality factor, implying that there are important individual differences in task-dependent lateralisation.

This is a very clearly written article, cleanly laid out and argued with good use of tables of figures.

The task main effect hypothesis (i.e. not all tasks are created equal from a laterality perspective) is a bit of a 'straw man', but I see the utility in the larger modelling scheme. I like the design and modelling approach used to address the question of task dependent individual differences (manifest in the comparison of the Person Effect Model versus the Task by Person Effect Model).

For Task F, I assume the button press was used to indicate whether participants thought the 'jabberwocky' was a sentence or not? This is not clear from the text.

In Figure 1, I don't understand why the data clouds in the right-hand column aren't centred on the cross-hairs defined by the red dashed lines (which I understand to show task means in the models). Shouldn't they be?

The justifications for the departures (e.g. sample, data handling) from the pre-registered approach are valid.

It is reassuring to see that LIs across repetitions of a given task are (at least relatively) stable.

Regarding the question of multivariate outliers (Fig 7), do the authors know/can they show that repeatability across time was comparable for the 'outlier' cases as for the non-outlier cases. In other words, are the outliers 'stably distinctive' in their profile, or is part of their distinctiveness their variability over time? Knowing that they are 'stably distinctive' would reinforce the take home message about individual differences.

In discussion: "A key question is whether blood flow measured using fTCD reflects the average of activity in a lateralised dorsal stream and a bilateral ventral stream, or whether the absolute dorsal stream activity is the main factor affecting the LI. In future we plan studies to address this question using fMRI." This is a good question; indeed, I wondered (I am just making an

observation here) whether fTCD was the best tool to answer the questions posed in this study (as opposed to say fMRI).

I agree with the statement: “The answer to the question of whether laterality is a unitary function is that, clearly, there are some individuals in whom laterality is different for different aspects of language. It is not, however, the case that there are two factors that act independently in the general population. Rather, the majority of people appear to have language laterality driven by a single process affecting all types of task, with a minority showing fractionation of language asymmetry.” This paragraph could do with some citations as this (individual cases who do not conform to the normal, uniformly left dominant picture) has been noted before.

Reviewer: 2

Comments to the Author(s)

Reviewed by Mohamed Seghier:

In this manuscript, Woodhead and colleagues used SEM to model between-task and between-subject variability in fTCD-based laterality indices. They showed that variability in laterality scores is meaningful and can be explained by task-by-subject factors.

The topic of this paper is very interesting; it is a nice addition to the current literature (debate?) about modelling/explaining the between-subject variability in lateralization. I have some minor comments about the current version of this manuscript and I hope the authors will find them useful.

As authors know, in addition to task and subjects factors, laterality also varies with brain region. However this spatial dimension is lost when using fTCD. So even if laterality scores or indices were stable across tasks, this does not necessarily mean that the spatial patterns were similar across tasks. ROI localization has strong impact on reliability and consistency between tasks, and many previous studies have shown significant between-region associations (spatial covariance) in lateralization at the cluster level (e.g. Seghier et al. 2011 HBM; Cai et al. 2010 Cereb Cortex) as well as at the voxel level (e.g. Seghier and Price, 2013 Brain Lang).

As acknowledged by the authors in Page 31, I don't think fTCD is suitable to test/compare predictions based on theoretical accounts that differ in terms of processing pathways (Hickok and Poeppel's dual route model of speech processing versus Dhanjal et al.'s lexical retrieval account). This section about theoretical accounts is not compelling in my opinion and I recommend to the authors to move it to the supplementary material (but I leave this to their discretion): (1) there is no strong rationale why only these two accounts have been selected among other previous but competing accounts; (2) the two accounts did not fully explain/predict the possible differences in laterality between tasks (e.g. see Task E); (3) the task where both accounts provided contrasting predictions was a task with poor test-retest reliability (Task A); (4) the RMSEA associated with both models (Table 2) was very poor which may suggest that those two models were 'poor' models given the fTCD data; (5) both models gave a worse fit than the Task Effect model; (6) there was heavy reliance on 'reverse inference' to generate predictions based on each account (e.g. which brain regions/streams were most likely implicated by a given task). Again, my point here is that this model comparison between predictions based on theoretical accounts would be an interesting question for future fMRI studies but not fTCD.

Did the authors select those six tasks from preliminary/piloting studies? It would be interesting for readers to know why you went for this set of tasks given the high level of variation and inconsistency between different tasks that you discussed in your 2017 systematic review (Peer J 2017). Based on that 2017 review, some useful recommendations were made about task design for

fMRI studies and I'm wondering whether those recommendations are also valid for optimal task design when using fTCD.

In the discussion section, please add a succinct paragraph about the usefulness of modeling laterality data with SEM: e.g. highlighting the kind of inferences one can make with SEM beyond what can already be inferred from Figures 5 and 6. Also, do you recommend SEM only for homogeneous samples given the significant impact of one left-hander (outliers?) on the original findings?

Page 5: how did you generate the synthetic data of Figure 1? Are these simulations based on the output of the selected models (similar to laterality simulations of Page 10)? Moreover, it would be useful for the reader to discuss how the empirical data (Figure 6) compared with your predictions from the simulated data (Figure 1). For instance, cross-task correlations were predicted for models with a 'Person' effect only (Figure 1), and real data showed significant cross-task correlations (Figure 6) that were better explained by a bifactor model.

The RMSEA values in Table 2 may indicate poor fit (except for the last model, all values are well above the upper limit of 0.08; see Hooper et al. 2008 "The Electronic Journal of Business Research Methods"). This could hint to a poor fit of the data covariance matrix or a kind of redundancy in the parametrization of the models. I recommend to the authors to succinctly mention this issue in the Discussion section, given current limitations in reporting RMSEA (and its interpretation) for studies with small degrees of freedom (see Kenny et al. 2015 "Sociological Methods & Research", 44(3):486-507).

It is puzzling why Task A (List Generation) showed poor test-retest reliability. In the behavioral analysis of Page 23, there was a significant session effect in the ANOVA analysis. Can this explain the poor reliability?

Task F (Syntactic Decision) showed weak lateralisation but still good test-retest reliability. I'm curious to know whether the absolute means (not the difference) of CBFV for left and right hemispheres were higher for Task F than Task A, which could maybe suggest higher blood flow velocity and hence robust signals during the execution of Task F.

Task F was not lateralized but showed high test-retest reliability. Recently, Wilson et al. (2017 NeuroImage: Clinical) found similar result with a different task: a picture naming task with high test-retest reproducibility despite showing no lateralization in fMRI. This can be mentioned in the Discussion section (Page 34).

In Table 1, please double check the t scores for the paired t-test comparisons between RTs. For example, Task F yielded identical means but the t score was 0.64.

Some CFI indices were negative in Table 2. CFI should range between 0.0 and 1.0.

Author's Response to Decision Letter for (RSOS-181801.R0)

See Appendix A.

Decision letter (RSOS-181801.R1)

12-Feb-2019

Dear Dr Woodhead,

I am pleased to inform you that your manuscript entitled "Testing the unitary theory of language lateralisation using functional transcranial Doppler sonography in adults" is now accepted for publication in Royal Society Open Science.

on behalf of Dr Shirley-Ann Rüschemeyer (Associate Editor) and Antonia Hamilton (Subject Editor)
openscience@royalsociety.org

Associate Editor Comments to Author (Dr Shirley-Ann Rüschemeyer):
Associate Editor
Comments to the Author:
Dear Dr. Woodhead,

Thank you for addressing the issues that were raised, in particular by Reviewer 2, in your revised submission. I am happy to accept the manuscript in its current state for publication in Royal Society Open Science.

Thank you for considering RSOS as an outlet for your work.

Best wishes,
Shirley-Ann Rueschemeyer

Appendix A

We thank the editor and referees for their many helpful comments. We respond to each in turn below and think that the manuscript is now considerably improved. For ease of reading, we have colour coded the comments in black, our responses in green, and any changes to the revised manuscript in red. The script used for the reanalyses described here are available on OSF (<https://osf.io/tkpm2/>).

1. Reviewer #1

1.1 Task F

For Task F, I assume the button press was used to indicate whether participants thought the 'jabberwocky' was a sentence or not? This is not clear from the text.

Thank you for spotting this omission. We have added this information to the Methods (p. 16):

"The participant was required to respond by button press following the '?' prompt to indicate whether they thought the sequence formed a sentence or not.

1.2 Figure 1

In Figure 1, I don't understand why the data clouds in the right-hand column aren't centred on the cross-hairs defined by the red dashed lines (which I understand to show task means in the models). Shouldn't they be?

Thank you for pointing this out – there was an error in the script generating the figure which meant the location of X and Y axes were transposed. We have now corrected Figure 1 so the red lines are centred on the means.

1.3 Outliers

Regarding the question of multivariate outliers (Fig 7), do the authors know/can they show that repeatability across time was comparable for the 'outlier' cases as for the non-outlier cases. In other words, are the outliers 'stably distinctive' in their profile, or is part of their distinctiveness their variability over time? Knowing that they are 'stably distinctive' would reinforce the take home message about individual differences.

We would like to thank the reviewer for this interesting suggestion. Ideally, we would investigate it by running the SEM analysis on the data from each session separately, but the design is not sufficiently well powered to do this. Instead, we have looked into the stability of the multivariate outlier participants in two different ways.

Firstly, we looked at each participant's change in LI values between session 1 and session 2, and summed the change across all six tasks. These data are plotted below. You can see that the outlier participants (in red) are no more variable over time than the other participants.

Secondly, we attempted to identify the multivariate outliers as shown in Figure 7, but keeping the data from Session 1 and Session 2 separate. We did this to get a rough impression of whether the same multivariate outliers would be identified in Session 1 and Session 2, but the analysis is not ideal, as the factor loadings were estimated in the SEM analysis using ALL of the data (i.e. from both sessions). Ideally, as mentioned above, the SEM would be repeated for Sessions 1 and 2 separately.

For each session, we followed the same procedure as in the original manuscript:

- The raw LI data for the six tasks at that session were weighted with the factor loadings from the SEM analysis
- The values for the six tasks were summed to give the factor scores for Factor 1 and Factor 2 for that session
- A linear regression model was fitted between the factor scores for Factor 1 and Factor 2 for that session, and Cook's distance was calculated for each participant.

This resulted in Cook's distance for each participant for Session 1 and Session 2. This data is plotted below. The original four outlier participants are marked in red. You can see that three of them (participants 10, 23 and 27) had relatively high Cook's distances at both sessions, indicating that they were stable multivariate outliers. By contrast, participant 29 was a strong outlier in Session 1, but not in Session 2.

We can tentatively conclude from this analysis that three out of four outliers were stable across sessions. We would prefer not to include this in the manuscript given the limitations of the statistical analysis, but these reviews will be published on RSOS and the analysis script is available on OSF (<https://osf.io/pu576/>) in case it is of interest to readers.

On performing this analysis, we noted an error in the Discussion, where it was stated that three out of the four outliers were left handers. In fact, all four outliers were left handed. We have now amended this in the Discussion (p. 35).

1.4 Choice of methodology

In discussion: “A key question is whether blood flow measured using fTCD reflects the average of activity in a lateralised dorsal stream and a bilateral ventral stream, or whether the absolute dorsal stream activity is the main factor affecting the LI. In future we plan studies to address this question using fMRI.” This is a good question; indeed, I wondered (I am just making an observation here) whether fTCD was the best tool to answer the questions posed in this study (as opposed to say fMRI).

This is a good question, and one often levelled against fTCD.

The spatial resolution of fTCD is certainly a limitation, but we would argue that it doesn’t diminish the validity of the findings presented in this manuscript. Please also see our response to Reviewer 2’s related comment (2.1) below.

Our approach is to capitalise on fTCD’s low cost in order to address suitable research questions with sample sizes that would be unaffordable in fMRI. By doing so, this will refine hypotheses and protocols that we can then employ in future fMRI studies.

1.5 Citations in discussion

I agree with the statement: “The answer to the question of whether laterality is a unitary function is that, clearly, there are some individuals in whom laterality is different for different aspects of language. It is not, however, the case that there are two factors that act independently in the general population. Rather, the majority of people appear to have language laterality driven by a single process affecting all types of task, with a minority showing fractionation of language asymmetry.” This paragraph could do with some citations as this (individual cases who do not conform to the normal, uniformly left dominant picture) has been noted before.

We have added the following sentence and citations to that paragraph (Discussion, p. 35):

“This is consistent with observations of discrepant hemispheric dominance for different language functions (1–3), which occurs in a small minority of participants.”

2. Reviewer #2

2.1 Spatial variability of lateralisation

As authors know, in addition to task and subjects factors, laterality also varies with brain region. However this spatial dimension is lost when using fTCD. So even if laterality scores or indices were stable across tasks, this does not necessarily mean that the spatial patterns were similar across tasks. ROI localization has strong impact on reliability and consistency between tasks, and many previous studies have shown significant between-region associations (spatial covariance) in lateralization at the cluster level (e.g. Seghier et al. 2011 HBM; Cai et al. 2010 Cereb Cortex) as well as at the voxel level (e.g. Seghier and Price, 2013 Brain Lang).

We agree that our results cannot speak to the spatial distribution of activation within the hemispheres, but we would argue that this does not diminish the validity of our findings. We are agnostic about which areas are activated – we accept that observing identical LI values for two tasks would not mean that they activate the same regions.

As fTCD measures changes in MCA blood flow velocity, it will be sensitive to activation anywhere within the MCA territory. Lateralisation indices calculated from fTCD data will therefore reflect the balance of activity from anywhere within the left and right MCA territories. In this respect, LI values in fTCD are roughly analogous to LI values from fMRI data within an MCA territory ROI taking into account activation extent as well as intensity. We must assume that fTCD is insensitive to activity outside of the MCA territory, which is potentially a limitation, but we would broadly expect the lateralised activity in these tasks to fall within MCA territory regions.

Hence, differences in mean LI values between tasks can be taken as evidence for differences in lateralisation within the MCA territory, regardless of precisely where that activity is. This is the basis of our first hypothesis.

With regards to our second hypothesis, covariance in LI values between tasks can be taken as evidence that individual differences in the amount of lateralised activity within MCA territory is consistent across the tasks. This may be driven by lateralised activity within the same regions, or by different regions that co-vary with each other. Either possibility would reflect a common cause of lateralisation.

Difference in spatial localisation of activity therefore does not undermine the validity of either hypothesis, or our interpretation of the results. However, we agree that it is necessary for the reader to be reminded of the spatial resolution of the technique, and so have highlighted this throughout the manuscript as follows:

“We used functional transcranial Doppler sonography to assess language lateralisation **within the middle cerebral artery territory** in 37 adults” (Abstract, p. 2)

“The fTCD data were used to derive laterality indices (LIs), which quantify the balance of activation in **the middle cerebral artery (MCA) territories** of the left and right hemispheres.” (Introduction, p. 3)

“Figure 5 shows clear differences from task to task in strength of cerebral lateralisation **within the MCA territory**” (Discussion, p. 32)

“LI as measured in fTCD reflects only the difference in blood flow within the left and right MCA territories, and is insensitive to activation outside of those areas.” (Discussion, p. 36)

2.2 Theoretical accounts / hypotheses

As acknowledged by the authors in Page 31, I don't think fTCD is suitable to test/compare predictions based on theoretical accounts that differ in terms of processing pathways (Hickok and Poeppel's dual route model of speech processing versus Dhanjal et al.'s lexical retrieval account). This section about theoretical accounts is not compelling in my opinion and I recommend to the authors to move it to the supplementary material (but I leave this to their discretion): (1) there is no strong rationale why only these two accounts have been selected among other previous but competing accounts; (2) the two accounts did not fully explain/predict the possible differences in laterality between tasks (e.g. see Task E); (3) the task where both accounts provided contrasting predictions was a task with poor test-retest reliability (Task A); (4) the RMSEA associated with both models (Table 2) was very poor which may suggest that those two models were 'poor' models given the fTCD data; (5) both models gave a worse fit than the Task Effect model; (6) there was heavy reliance on 'reverse inference' to generate predictions based on each account (e.g. which brain regions/streams were most likely implicated by a given task). Again, my point here is that this model comparison between predictions based on theoretical accounts would be an interesting question for future fMRI studies but not fTCD.

We prefer to retain the tests of models of means, because the paper is structured around the sequential testing of the models in Figure 1. We feel that, though our study results did not successfully distinguish between models, the approach is worth presenting in full. Previous literature has tended to conflate task variation and individual differences in laterality, and we hope this illustration will at least clarify that they make independent predictions.

Regarding points 4-5, we noted in the account of the models that we would expect poor fit of any means model, because there is substantial variance due to individual differences which is not taken into consideration. It is relative rather than absolute fit that is of interest in this context.

Having said that, we agree with the other points made here, but we think this highlights the need to devise better ways of testing models of processing pathways. Our rather simple approach was not successful, and we discuss some of the limitations in our Discussion. But what kind of evidence would distinguish the Hickok/Poeppel vs Dhanjal models, let alone the numerous other models that the reviewer alludes to? Evidence from fMRI would help us distinguish different pathways involved in task performance, but we still would have to confront the difficulties of devising 'pure' tasks, and the lack of specification of what counts as significant lateralisation.

2.3 Choice of tasks

Did the authors select those six tasks from preliminary/piloting studies? It would be interesting for readers to know why you went for this set of tasks given the high level of variation and inconsistency between different tasks that you discussed in your 2017 systematic review (Peer J 2017). Based on that 2017 review, some useful recommendations were made about task design for fMRI studies and I'm wondering whether those recommendations are also valid for optimal task design when using fTCD.

Yes, considerable thought went into the design of the tasks, which was omitted from the manuscript for simplicity's sake. We agree with the reviewer that including some background would be relevant for the reader.

Guided by the recommendations in our previous review (6) we had two main aims in designing the language tasks: 1) to assess lateralisation across a broad range of language functions; and 2) to keep the tasks as closely matched for non-linguistic demands (e.g. timings, input / output modality, task difficulty) as possible. The number of tasks was limited by how many tasks we could test comfortably in one session. We identified phonology, semantics, syntax as three key language domains, and speech production and comprehension as key language functions. Reading and writing were considered as alternative functions, but they engage other neural systems specific to written language, and so would complicate interpretation. We were also mindful of minimising literacy requirements, as we intended to use the tasks in a dyslexic participant group in a future study.

To choose the specific tasks for each language domain / function, we searched the literature for established tasks, preferably ones that had shown good test-retest reliability in either fMRI or fTCD, although there was very limited reliability data available. The speech production tasks (Sentence Generation and List Generation) were based on the tasks used for the BIL&GIN database (7,8). The Phonological Decision task was based on the fTCD study by Payne and colleagues (9), and the Semantic Decision task was designed to match. We couldn't find an established speech comprehension task that had an active decision component but without requiring a spoken response; hence we designed the Sentence Comprehension task using existing pictures (from the TROG) and spoken sentences that were relatively taxing for adult participants to comprehend. Finally, for syntactic processing, we aimed to find a task that did not also tap sentential semantic processing. We chose the Jabberwocky stimuli (10) as a means of manipulating syntactic structure without any meaningful content.

In piloting we soon found that it was impossible for the different language tasks to be perfectly matched, so a balance had to be struck. The number of trials per condition were the same for all tasks. We manipulated the task timings so that trial lengths (and therefore the time on task and time since baseline) would be the same for all conditions. We were able to make the timings within a trial for the phonological decision, semantic decision and sentence comprehension tasks identical. Pilot testing was carried out and difficulty levels were titrated to make task performance (accuracy and RT for decision tasks; number of words spoken for speech production tasks) as similar as possible between tasks.

We have added the following details to the manuscript:

“The tasks were designed to assess lateralization across a broad range of language functions (namely phonology, semantics, syntax, speech production and speech perception), whilst keeping non-linguistic demands as closely matched as possible.” (Introduction, p. 3)

“The number of trials per condition were the same for all tasks. We manipulated the task timings so that trial lengths (and therefore the time on task and time since baseline) would be the same for all conditions. We were able to make the timings within a trial for the phonological decision, semantic decision and sentence comprehension tasks identical. Pilot testing was carried out and difficulty levels were titrated to make task performance (accuracy and RT for decision tasks; number of words spoken for speech production tasks) as similar as possible between tasks.” (Methods, p. 11)

2.4 Usefulness of SEM

In the discussion section, please add a succinct paragraph about the usefulness of modeling laterality data with SEM: e.g. highlighting the kind of inferences one can make with SEM beyond what can

already be inferred from Figures 5 and 6. Also, do you recommend SEM only for homogeneous samples given the significant impact of one left-hander (outliers?) on the original findings?

We have added some wording explaining the motivation for using SEM in both the Introduction (p. 7) as well as the Discussion (p. 32):

“SEM was used for a number of reasons, including: 1) it allowed us to model the unobserved factors (also known as latent variables or constructs) predicting lateralisation strength across the observed tasks and sessions (also known as manifest variables or indicators); 2) it modelled both the mean strength of lateralisation for each task as well as the covariance between tasks; 3) it explicitly modelled the residuals associated with each latent variable, which allowed measurement error to be accounted for; and 4) it allowed for different models to be compared directly using likelihood ratio hypothesis testing.” (p. 7)

“The SEM analyses provided an economical approach for considering a range of hypotheses within a single framework, allowing us to test statistically whether this pattern between tasks was significant. SEM was used as it allowed us to formalise our hypotheses in statistical models that could then be compared using likelihood ratio significance tests, rather than making subjective inferences from the pattern of means and correlations alone.” (p. 32)

However, our experience with SEM has taught us that the method needs to be used with caution. We still think it is useful, as investigating reasons for contradictory results can be very illuminating. If we had not attempted to use SEM and had just looked at correlation patterns, we would have missed the heterogeneity of the sample. We now add a note to that effect (Discussion, p. 35):

“These results indicate that SEM can be useful for studying individual variation in patterns of laterality, but it needs to be interpreted with caution. It can be tempting to focus on achieving a good model fit, but, as Hooper et al (46) noted, the purpose of SEM should be model-testing, and even a well-fitting model may be sub-optimal. Our data illustrate that point clearly: having found support for a bifactor model with no paths between the latent factors, we expected that we would have two independent factors that were binormally distributed. By performing further checks in terms of a leave-one-out analysis and visualisation of the two extracted factors, it became clear that this was not the case, and that our data contained a mixture of two subgroups. On the one hand, we can conclude that SEM is not optimal for investigating multivariate aspects of laterality, because it assumes multivariate normality. On the other hand, by applying SEM and studying the anomalies that resulted, we gained insights into the heterogeneous nature of our sample.”

2.5 Simulated data

Page 5: how did you generate the synthetic data of Figure 1? Are these simulations based on the output of the selected models (similar to laterality simulations of Page 10)? Moreover, it would be useful for the reader to discuss how the empirical data (Figure 6) compared with your predictions from the simulated data (Figure 1). For instance, cross-task correlations were predicted for models with a ‘Person’ effect only (Figure 1), and real data showed significant cross-task correlations (Figure 6) that were better explained by a bifactor model.

The synthetic data was simulated using multilevel models described in the Introduction, not the SEM models themselves. This was done as a simple and intuitive way to visualise the impact of different factors on the data structure and to motivate the hypotheses. However, the power simulations did

use the SEM models. We will provide a more thorough description of these simulations in an R Markdown document on OSF.

To clarify, the simulation for the 'Person effect' model predicted that if there is only one latent lateralisation factor, all cross-task correlations would be strong. Also, importantly, if there was a bifactor structure (i.e. 'Task by Person effect'), tasks that loaded onto the same latent factor would still have strong cross-task correlations, but tasks that loaded onto different factors would have weak correlations. Figure 6, and the results of the SEM, support the latter – there is a cluster of tasks that correlate strongly together, but weakly with other tasks (most notably task F). We have expressed this more clearly in the Discussion as follows (p. 31):

“This pattern of correlations is indicative of a Person by Task interaction (as hypothesised in Figure 1): there is a cluster of tasks (B, C, D and E) that have strong cross-task correlations, indicating that they are driven by a common lateralising factor. In contrast, Task F has weak correlations with these tasks, despite having high test-retest reliability. This suggests that lateralisation on this task is driven by a second, independent factor.”

2.6 RMSEA values

The RMSEA values in Table 2 may indicate poor fit (except for the last model, all values are well above the upper limit of 0.08; see Hooper et al. 2008 “The Electronic Journal of Business Research Methods”). This could hint to a poor fit of the data covariance matrix or a kind of redundancy in the parametrization of the models. I recommend to the authors to succinctly mention this issue in the Discussion section, given current limitations in reporting RMSEA (and its interpretation) for studies with small degrees of freedom (see Kenny et al. 2015 “Sociological Methods & Research”, 44(3):486-507).

Our approach was to use SEM to test a series of pre-specified, hypothesis-driven models. This differs from using SEM to identify an optimal fit to the data via posthoc model modifications. We fitted all models without modification from our preregistered analysis plan and, to this end, we have reported the fit of every model, even suboptimal ones, as the important detail is whether they offer an improved fit relative to previous models in the series. This approach is of particular importance in designs with (relatively) small sample sizes and low degrees of freedom, where overfitting is a real concern.

We acknowledge that the simpler models (i.e. the models of means) had poor fit but they were not necessarily required to fit the data well as they were likely an over-simplification of the underlying structure. We only sought to discount them as viable, more parsimonious models than the later complex models. The poor model fit limits their interpretability; but our intention was not to interpret them in isolation. Rather, we view them as stepping stones along the way, by identifying the optimal structure to explain the LI means prior to adding the covariance structure. Only the later models achieved satisfactory fit for interpretation.

We have added the following text to the Introduction to clarify our approach to SEM (p. 9):

“Our approach was to use SEM to test a series of pre-specified, hypothesis-driven models. This differs from using SEM to identify an optimal fit to the data via posthoc model modifications. We fitted all models without modification from our preregistered analysis plan and, to this end, we have reported the fit of every model, even suboptimal ones, as the important detail is whether they offer an improved fit relative to previous models in the

series. This approach is of particular importance in designs with (relatively) small sample sizes and low degrees of freedom, where overfitting is a real concern.”

We also acknowledge the poor model fit of the simpler models in the Discussion (p. 34):

“We acknowledge that the SEM models of means had poor fit but they were not required to fit the data well as they were likely an over-simplification of the underlying structure. We only sought to discount them as viable, more parsimonious models than the later complex models. The poor model fit limits their interpretability; but our intention was not to interpret them in isolation. Rather, we view them as stepping stones along the way, by identifying the optimal structure to explain the LI means prior to adding the covariance structure. Only the later models achieved satisfactory fit for interpretation.”

2.7 Test-retest reliability of Task A

It is puzzling why Task A (List Generation) showed poor test-retest reliability. In the behavioral analysis of Page 23, there was a significant session effect in the ANOVA analysis. Can this explain the poor reliability?

I think it's unlikely. The behavioural measure for Task A was the number of words produced. There was only a small change in this measure between sessions (session 1 = 9.5, session 2 = 9.6), and this did not correlate with the change in LI between sessions ($R=-0.13$, $p=0.436$).

It seems more likely to me that the problem with Task A is that there was no way of controlling / judging what participants were doing during the covert generation period (from which LI was calculated). Participants could perform the overt task perfectly without actually doing the covert rehearsal at all.

We have added a note about this issue to the Discussion (p. 33-34):

“The question of why task A (List Generation) had such low test-retest reliability remains open, but it is possible that the covert generation of speech sequences was not sufficiently engaging to elicit robust, reliable brain activity. The behavioural measure used for this task (number of words spoken) is not a very useful index of engagement in the task, as the overt speech generation could be performed perfectly even if the participant did not covertly generate the words at all. This is in contrast to task D (Sentence Generation) where covert generation would aid the participant's overt sentence reporting.”

2.8 Blood flow velocity in tasks A and F

Task F (Syntactic Decision) showed weak lateralisation but still good test-retest reliability. I'm curious to know whether the absolute means (not the difference) of CBFV for left and right hemispheres were higher for Task F than Task A, which could maybe suggest higher blood flow velocity and hence robust signals during the execution of Task F.

The plot below shows the normalised CBFV in left and right sensors for tasks A and F, averaged across all participants and both sessions. The two tasks had different time-courses, but the start of the period of interest for both was 6 seconds (indicated by a vertical line). For task A (List Generation), the overt response began at 17 seconds, where a peak in activity can be seen. The period of interest ended at 17 seconds for this reason.

Looking between 6 and 17 seconds only, CBFV was higher for task F than task A. This difference was significant in both the left hemisphere ($t(73)=2.00, p=.049$) and the right hemisphere ($t(73)=4.21, p<.001$).

As mentioned above (item 2.6) we have added an additional note to the Discussion (p.32) highlighting the possibility that weaker signals in task A may have contributed to its poor test-retest reliability. The script used to perform this analysis is available on OSF (<https://osf.io/pu576/>).

2.9 Test-retest reliability of Task F

Task F was not lateralized but showed high test-retest reliability. Recently, Wilson et al. (2017 NeuroImage: Clinical) found similar result with a different task: a picture naming task with high test-retest reproducibility despite showing no lateralization in fMRI. This can be mentioned in the Discussion section (Page 34).

Thank you for directing us to this very relevant paper. We have included a citation to it in the Discussion as follows (p. 36):

“A similar finding was recently reported in an fMRI study investigating the validity and reliability of different language paradigms, which showed that a picture naming task showed high test-retest reliability despite not being lateralised (11). These findings demonstrate that strength as well as direction of lateralisation for a task are both stable traits.”

2.10 Table 1

In Table 1, please double check the t scores for the paired t-test comparisons between RTs. For example, Task F yielded identical means but the t score was 0.64.

We have double checked these values and they are correct. In the case of Task F, the mean RT for Session 1 was 0.334 and for Session 2 it was 0.329, so when rounded they appeared to be identical. We have changed these values to three decimal places in the manuscript (Table 1) to avoid confusion.

Measure	Session	Task B	Task C	Task E	Task F
Accuracy (%)	1	91.3 (5.55)	95.9 (3.08)	92.5 (4.81)	89.6 (8.31)
	2	93.3 (4.28)	95.0 (3.06)	94.2 (3.79)	89.4 (8.28)
	1 vs 2	t=-3.27, p=.002	t=1.61, p=.115	t=-2.70, p=.011	t=-0.07, p=.944
Reaction times (s)	1	1.66 (0.22)	1.14 (0.2)	2.17 (0.12)	0.334 (0.08)
	2	1.49 (0.21)	1.06 (0.2)	2.11 (0.15)	0.329 (0.07)
	1 vs 2	t=8.73, p<.001	t=4.77, p<.001	t=3.27, p=.002	t=0.64, p=.528
Omitted responses (%)	1	2.34	0.84	2.79	4.20
	2	0.78	0.60	1.62	4.44

2.11 CFI values

Some CFI indices were negative in Table 2. CFI should range between 0.0 and 1.0.

Thank you for pointing out this error. We realise now that the CFI values should be truncated if less than zero or greater than one (e.g. <http://www.davidakenny.net/cm/fit.htm>), and have updated Table 2 accordingly.

Model	Description	-2LogL	df	BIC	CFI	RMSEA	Chi Square test Compared to	p
Fully Saturated Model	Free means and variances	1574.4	411	90.4	NA	NA	-	NA
Task Effect Model	Stable means and variances	1580.8	423	53.4	0.022	0.292	Fully Saturated Model	0.896
Population Bias Model	Equal means and variances	1715.5	433	151.9	0	0.337	Task Effect Model	<0.001
Dorsal Stream Model	Means for tasks AB > DEF > C	1664.8	429	115.7	0	0.323	Task Effect Model	<0.001
Lexical Retrieval Model	Means for tasks BD > ACF	1631.6	429	82.5	0	0.306	Task Effect Model	<0.001
Person Effect Model	Covariances have one factor structure	1378.4	417	-127.4	0.805	0.136	Task Effect Model	<0.001
Task x Person Effect Model	Covariances have bifactor structure	1337.8	412	-149.9	0.947	0.073	Person Effect Model	<0.001

References

1. Gaillard WD, Balsamo L, Xu B, McKinney C, Papero PH, Weinstein S, et al. fMRI language task panel improves determination of language dominance. *Neurology*. 2004 Oct;63(8):1403–8.
2. Stroobant N, Buijs D, Vingerhoets G. Variation in brain lateralization during various language tasks: A functional transcranial Doppler study. *Behav Brain Res*. 2009;199(2):190–6.
3. Tailby C, Abbott DF, Jackson GD. The diminishing dominance of the dominant hemisphere: Language fMRI in focal epilepsy. *NeuroImage Clin [Internet]*. 2017;14:141–50. Available from: <http://dx.doi.org/10.1016/j.nicl.2017.01.011>

4. van der Zwan A, Hillen B, Tulleken CAF, Dujovny M, Dragovic L. Variability of the territories of the major cerebral arteries. *J Neurosurg* [Internet]. 1992;77:927–40. Available from: <https://thejns.org/view/journals/j-neurosurg/77/6/article-p927.xml>
5. van der Zwan A, Hillen B. Review of the variability of the territories of the major cerebral arteries. *Stroke*. 1991;22(8):1078–84.
6. Bradshaw AR, Thompson PA, Wilson AC, Bishop DVM, Woodhead Z. Measuring language lateralisation with different language tasks: a systematic review. *PeerJ* [Internet]. 2017;5:e3929. Available from: <https://peerj.com/articles/3929>
7. Mazoyer B, Mellet E, Perchey G, Zago L, Crivello F, Jobard G, et al. BIL&GIN: A neuroimaging, cognitive, behavioral, and genetic database for the study of human brain lateralization. *Neuroimage* [Internet]. 2016;124:1225–31. Available from: <http://dx.doi.org/10.1016/j.neuroimage.2015.02.071>
8. Mazoyer B, Zago L, Jobard GG, Crivello F, Joliot M, Perchey G, et al. Gaussian mixture modeling of hemispheric lateralization for language in a large sample of healthy individuals balanced for handedness. *PLoS One*. 2014 Jun 30;9(6):9–14.
9. Payne H, Gutierrez-Sigut E, Subik J, Woll B, MacSweeney M. Stimulus rate increases lateralisation in linguistic and non-linguistic tasks measured by functional transcranial Doppler sonography. *Neuropsychologia* [Internet]. 2015;72:59–69. Available from: <http://dx.doi.org/10.1016/j.neuropsychologia.2015.04.019>
10. Fedorenko E, Nieto-Castañón A, Kanwisher N. Lexical and syntactic representations in the brain: An fMRI investigation with multi-voxel pattern analyses. *Neuropsychologia* [Internet]. 2012 Mar;50(4):499–513. Available from: <http://linkinghub.elsevier.com/retrieve/pii/S0028393211004210>
11. Wilson SM, Bautista A, Yen M, Lauderdale S, Eriksson DK. Validity and reliability of four language mapping paradigms. *NeuroImage Clin* [Internet]. 2017;16:399–408. Available from: <https://doi.org/10.1016/j.nicl.2016.03.015>